# Constraints on global oceanic emissions of $N_2O$ from observations and models

Erik T. Buitenhuis[1,2], Parvadha Suntharalingam[1], and Corinne Le Quéré[1,2]

[1]School of Environmental Sciences, University of East Anglia, Norwich, United Kingdom
[2]Tyndall Centre for Climate Change Research, University of East Anglia, Norwich, United Kingdom

*Correspondence to:* Erik T. Buitenhuis (E-mail: http://greenocean-data.uea.ac.uk/.feedback.html)

**Abstract.** We estimate the global ocean $N_2O$ flux to the atmosphere and its confidence interval using a statistical method based on model perturbation simulations and their fit to a database of $\Delta pN_2O$ (n=6136). We evaluate two submodels of $N_2O$ production. The first submodel splits $N_2O$ production into oxic and hypoxic pathways following previous publications. The second submodel explicitly represents the redox transformations of N that lead to $N_2O$ production (nitrification and hypoxic denitrification) and $N_2O$ consumption (suboxic denitrification), and is presented here for the first time. We perturb both submodels by modifying the key parameters of the $N_2O$ cycling pathways (nitrification rates, $NH_4^+$ uptake, $N_2O$ yields under oxic, hypoxic and suboxic conditions), and determine a set of optimal model parameters by minimisation of a cost function against 4 databases of N cycle observations derived from observed and model $\Delta pN_2O$ concentrations. Our estimate of the global oceanic $N_2O$ flux resulting from this cost function minimisation is 2.4 ± 0.8 and 2.5 ± 0.8 Tg N $y^{-1}$ for the 2 $N_2O$ submodels. These estimates suggest that the currently available observational data of surface $\Delta pN_2O$ constrain the global $N_2O$ flux to a narrower range relative to the large range of results presented in the latest IPCC report.

## 1 Introduction

Nitrous oxide ($N_2O$) is the third most important contributor to anthropogenic radiative forcing, after carbon dioxide ($CO_2$) and methane ($CH_4$) (Myhre et al., 2013). It is also currently estimated as the dominant contributor to stratospheric ozone depletion (Portmann et al., 2012). Yet our quantitative understanding of the magnitude and processes controlling natural $N_2O$ emissions from the Earth surface to the atmosphere is very poor. A range of methods have been used to constrain total oceanic $N_2O$ emissions, including the combination of surface ocean $N_2O$ partial pressure anomalies with gas-exchange parameterizations (Nevison et al., 1995), empirically derived functional relationships applied to global ocean datasets (Nevison et al., 2003; Freing et al., 2012), and ocean biogeochemistry models (Suntharalingam and Sarmiento, 2000; Suntharalingam et al., 2000; Jin and Gruber, 2003; Martinez-Rey et al., 2015). In spite of the multiple methods used, the reported oceanic emissions of $N_2O$ is still poorly constrained, ranging from 1.9 to 9.4 Tg N $y^{-1}$ according to the latest report of the Intergovernmental Panel on Climate Change (IPCC Ciais et al., 2013). The uncertainty in the oceanic emissions of $N_2O$ accounts for a large part of the total uncertainty in the natural $N_2O$ emissions, which are approximately 11 Tg N $y^{-1}$ (Ciais et al., 2013). Part of the uncertainty in the oceanic emissions is whether estuaries are included, which could emit as much as 2.3 - 3.6 Tg N $y^{-1}$ (Bange et al., 1996).

The large uncertainty in the oceanic emissions of N$_2$O stems from the complexity of its production pathways. There are two main pathways of N$_2$O production in the ocean, nitrification and denitrification, which both stem from redox reactions of nitrogen, under oxic and hypoxic conditions, respectively (Fig. 1). N$_2$O is formed as a byproduct of marine nitrification of ammonium (NH$_4^+$) to nitrate (NO$_3^-$); N$_2$O is also an intermediate product of denitrification, during the reduction of NO$_3^-$ to

nitrogen gas (N$_2$) (Frame and Casciotti, 2010; Loescher et al., 2012; Merbt et al., 2012). Denitrification can also consume N$_2$O, using extracellular N$_2$O, and reduce it to N$_2$ (Bange, 2008). In the oxic part of the ocean (i.e. most of the ocean, 97% >34 $\mu$mol O$_2$ L$^{-1}$ (using O$_2$ data taken from Bianchi et al., 2012)) denitrification is suppressed, and the primary formation pathway is usually ascribed to nitrification (Cohen and Gordon, 1978), although denitrification may be significant in the anaerobic centres of large marine snow particles in oxic waters (Klawonn et al., 2015). Oceanic N$_2$O production in oxic regions is often derived

from the linear relationships observed between apparent oxygen utilization (AOU) and apparent N$_2$O production ($\Delta$N$_2$O) (e.g. Yoshinari, 1976; Cohen and Gordon, 1978). However, the $\Delta$N$_2$O/AOU ratio varies in different water masses and oceanic regions (Suntharalingam and Sarmiento, 2000). Previous studies have suggested that differences in the $\Delta$N$_2$O/AOU ratio could be driven by changing N$_2$O yields under varying pressure and temperature (Butler et al., 1989) or varying O$_2$ concentration (Nevison et al., 2003). Additional mechanisms not yet quantified could include variations in the elemental stoichiometry

of the organic matter that is being remineralised, and spatial separation of organic matter remineralisation and nitrification. Throughout the manuscript we will refer to N$_2$O stoichiometries relative to O$_2$, NH$_4^+$ and NO$_3^-$ as ratios, because they have been optimised against global databases of concentration measurements, rather than from microbiological yields. Using the latter would be more mechanistically satisfying, but the relevant yields are at present insufficiently constrained by observations.

Estimates of the contribution from suboxic regions of the ocean (about 3%) to the global N$_2$O flux vary from net depletion

via denitrification (Cohen and Gordon, 1978), to 33% for the total N$_2$O production in the suboxic ocean (Suntharalingam et al., 2012), and to more than 50% from denitrification alone (Yoshida et al., 1989). This ambiguity remains unresolved. Bottom-up microbial physiology data is relatively scarce (see Sect. 2.4 - 2.6), while top-down data needs relatively complicated inverse methods to estimate the contribution from suboxic regions. These inverse methods are complicated both because of the variation in the $\Delta$N$_2$O/AOU ratio, which is negative under suboxic conditions, maximal under hypoxic conditions and lower under oxic

conditions (e.g. 0.31 - 0.033 mmol/mol, Law and Owens, 1990), and because the influence of mixing gradients make in situ ratios an unreliable gauge to the biological yields under in situ conditions (Nevison et al., 2003).

Here, we estimate the global ocean N$_2$O flux to the atmosphere and its confidence interval. First, we estimate N$_2$O flux from observations only (Sect. 2.1). This estimate has large uncertainty. We subsequently use a statistical approach introduced by Buitenhuis et al. (2013a) to estimate the global oceanic emissions of N$_2$O and its confidence interval by combining ocean

N$_2$O model simulations with a global database of measurements of surface $\Delta$pN$_2$O. This approach involves minimisation of a cost function that compares a series of model simulations with a global database of point measurements of surface $\Delta$pN$_2$O. To achieve this, we use 4 observational databases of the N cycle (Sect. 2.2) to extend the global ocean biogeochemistry model PlankTOM10 (Le Quere et al., 2016b) with additional N cycle processes. We derive the biogeochemical parameters for nitrification rate and phytoplankton use of NH$_4^+$ from the observational databases of nitrification rate and NH$_4^+$ concentration

(databases (1) and (2) and Sect. 2.4-2.5). Then, we describe two separate submodels of different levels of complexity that

represent $N_2O$ cycling pathways (Sect. 2.6-2.7). Finally, we apply the statistical approach (Sect. 2.8) to the two submodels to estimate the $N_2O$ production in the low $O_2$ regions from the depth resolved $N_2O$ concentration database (database (3) and Sect. 3.1), and the global oceanic $N_2O$ flux from the surface $\Delta pN_2O$ database (database (4) and Sect. 3.2), followed by a discussion of the results (Sect. 4).

## 2   Ocean N cycle

### 2.1   Calculation of global ocean $N_2O$ production from N cycle observations

In this section we provide an initial estimate of global marine $N_2O$ production based on observationally derived quantities characterising marine productivity and the global ocean N cycle. This follows a similar method to Cohen and Gordon (1979), who estimated ocean $N_2O$ production using Redfield type ratios. $N_2O$ is produced either during production of $NO_3^-$ in $NH_4^+$ oxidation or during $NO_3^-$ reduction in denitrification (Fig. 1). We therefore base the $N_2O$ production on total $NO_3^-$ turnover, calculated from primary production times the f-ratio. The f-ratio is the fraction of primary production that is supported by nitrate. Primary production (PP) was estimated at $58 \pm 7$ Pg C $y^{-1}$ based on $^{14}C$ primary production measurements (n=50,050), parameter perturbations of a previous version of the model uses here, and Eq. 5 (Buitenhuis et al., 2013a). We compiled a database of uptake rates of $NO_3^-$, $NH_4^+$ and urea, which gives an average f-ratio of $0.29 \pm 0.18$ (Fig. 2, large symbols, n=34). The globally averaged $\Delta N_2O$/AOU ratio was calculated from the MEMENTO database (Bange et al., 2009) as $81.5 \pm 1.4$ $\mu$mol/mol (Fig. 3). Finally, since primary production is expressed in carbon terms, and $N_2O$ production was correlated with oxygen ($O_2$) utilization, we need to include the -$O_2$:C ratio (the - sign indicates the $O_2$ is consumed as $CO_2$ is produced), which was taken from Anderson and Sarmiento (1994) as $170 \pm 10$ / $117 \pm 14$, and the molar weights of C (12) and N in $N_2O$ (28). Here and in the rest of the paper, errors were progagated in the usual way:

$$error = \sqrt{(\frac{error\,of\,A}{A})^2 + (\frac{error\,of\,B}{B})^2 + ...} * A * B * ... \tag{1}$$

Thus $N_2O$ production was calculated as PP *f-ratio*-$O_2$:C *$\Delta N_2O$/AOU. Our best estimate of $N_2O$ production using this method is $58 \pm 7$ *1000 *$0.29 \pm 0.18$ *$170 \pm 10$ /$117 \pm 14$ *81.5e-6 $\pm$ 1.4e-6 *28 /12 = $4.6 \pm 3.1$ Tg N $y^{-1}$. This estimate lies in the middle of other reported estimates (Fig. 4) but the 68% confidence interval is very large. We therefore investigate the $N_2O$ fluxes using a model optimised with observations in the rest of the paper.

### 2.2   Observational databases for model development

We used four databases to tune or optimise different aspects of the N cycle in the PlankTOM10 ocean biogeochemistry model. The number of datapoints reported for each database are after gridding to $1° \times 1° \times 12$ months $\times$ 33 depths (World Ocean Atlas 2009). The databases used are (1) $NH_4^+$ specific nitrification rate ($d^{-1}$, raw data n=425, gridded data n=296) as described in Yool et al. (2007); (2) surface $NH_4^+$ concentration distribution ($\mu$mol $L^{-1}$, raw data n=33079, gridded data n=2343) that combines the dataset used in Paulot et al. (2015) with data held by the British Oceanographic Data Centre in January 2014

(Johnson et al. in prep., http://www.bodc.ac.uk); (3) depth-resolved $N_2O$ concentration from the MEMENTO project (nmol $L^{-1}$, https://memento.geomar.de/; Bange et al., 2009, ; downloaded 4 June 2014, raw data n=14342, gridded data n=8047); and (4) surface partial pressure of $N_2O$ ($pN_2O$) also from MEMENTO (ppb, downloaded 16 Sept. 2015, raw data n=227463, gridded data n=6136). Since there is at present no formal quality control beyond that performed by individual contributors to

the MEMENTO database and a check by the database administrators that the values make physical sense (Kock and Bange, 2015), we have taken the database at face value. $pN_2O$ was converted to $\Delta pN_2O$ using atmospheric $pN_2O$:

$$pN_2O_{atm} = 0.000009471353 \times Y^3 - 0.052147139 \times Y^2 + 95.68066 \times Y - 58228.41 \tag{2}$$

(A. Freing, pers. comm., correction to Freing et al., 2009), in which Y is the decimal year. The average absolute difference relative to the global average $pN_2O_{atm}$ data from the NOAA/ESRL Global Monitoring Division (ftp://ftp.cmdl.noaa.gov/hats/

n2o/combined/HATS_global_N2O.txt) is 0.5 ppb between 1977 and 2014 and 0.3 ppb between 2000 and 2014.

## 2.3   Cost Function Formulation

To parameterise the model N cycle, we use a cost function to minimize the difference between model and observations, following the methods of Buitenhuis et al. (2013a):

$$cost function = 10^{\Sigma|log_{10}(model/observation)|/n} \tag{3}$$

This formulation gives equal weight to the relative correspondence between model and observations at small and large observational values. A value of 2 means that, on average, the model deviates from the observations by a factor 2 in either direction. To calculate the cost function (and also to calculate MSE in Eq. 6), the model was regridded to the same grid as the observations, and residuals were calculated at months and places where there are observations. The cost function results for the optimised simulations are summarised in Table 1.

## 2.4   Nitrification

Our initial biogeochemical model configuration is PlankTOM10 (Le Quere et al., 2016b), which represents growth and loss terms from ten Plankton Functional Types (PFTs), including $N_2$-fixers, picoheterotrophs (*Bacteria* plus *Archaea*) and denitrification rate, but not denitrifier biomass. A full model description and parameter values are provided in the supplementary material. Here, we extend the model representation of redox reactions in the N cycle, to create the global biogeochemical

model PlankTOM10.2. We describe the new N cycle components below.

In order to represent nitrification rate, the state variable for dissolved inorganic nitrogen was split into $NO_3^-$ and $NH_4^+$. Respiration by all PFTs produces $NH_4^+$. The parameterization for nitrification used in our model is based on the analysis of a database of $NH_4^+$-specific nitrification rates (Yool et al., 2007). Yool et al. (2007) found that observed nitrification rates are highly variable, with no obvious relationship with either latitude or depth. In their model they therefore used a constant rate of

0.2 $d^{-1}$ throughout the ocean. Implementing this rate in our model resulted in a cost function relative to the nitrification rate observations of 4.22 (Table 1). We tested if including temperature, $O_2$ or light dependence improves the ability of the model to

reproduce observed nitrification rates. Regarding the response of ammonia oxidizing *Archaea* (AOA), the main nitrifiers in the ocean (Francis et al., 2005; Wuchter et al., 2006; Loescher et al., 2012), to temperature, we are only aware of the measurements of Qin et al. (2014). These show a ~4-fold variation in maximum growth rate between 3 strains, which poorly constrains the temperature dependence of AOA. We therefore first used a generic $Q_{10}$ of 2 and optimised the rate at 0°C using the nitrification rate observations. This led to only a slightly improved representation of the observations (cost function = 4.18). Although the response of AOA and ammonia oxidizing *Bacteria* (AOB) to $O_2$ has only been measured at 21-25 °C (Frame and Casciotti, 2010; Loescher et al., 2012), which limits the range of $O_2$ concentrations, there was a significant logarithmic relationship between $N_2O$ yield and $O_2$ (Fig. 5). A logarithmic function fit the data better than linear, exponential or power functions. Since nitrification consumes $O_2$, in the model it decreases as remineralisation switches from $O_2$ to $NO_3$ (supplementary material Eq. 70, 61, 67). Implementing this response to $O_2$ led to only a further small improvement of the model nitrification rate relative to the observations (cost=4.16). This implies that nitrification never becomes $O_2$ limited, reflecting a lack of data to parameterise an expected decrease. As will be described more fully in Sect. 3.1, we used observed $O_2$ concentrations in the simulations (Bianchi et al., 2012) rather than interactively modelled $O_2$, to minimise the impact of model biases in simulated $O_2$ fields (Suntharalingam et al., 2012). The response of AOA to light is estimated to be 50% inhibited at 5 $\mu$mol photons m$^{-2}$ s$^{-1}$. However, this estimate is not well constrained (Merbt et al., 2012). Implementing this light response did not improve the model, either in combination with the $O_2$ and temperature responses or with the temperature response only, and was subsequently omitted. The lack of improvement in nitrification rates by adding light inhibition might reflect the lower sensitivity of AOA to light found by Qin et al. (2014).

## 2.5  Phytoplankton $K_{1/2}$ for $NH_4^+$ uptake

We used the calculation of the preferential uptake of $NH_4^+$ over $NO_3^-$ by phytoplankton PFTs of Vallina and Le Quere (2008)(supplementary material Eq. 9). The $K_{1/2}$ of phytoplankton for $NH_4^+$ has mostly been measured based on uptake rates (syntheses by Goldman and PM, 1983; Killberg-Thoreson et al., 2014). Aksnes and Egge (1991) have shown a theoretical expectation of a linear increase of $K_{1/2}$ with cell radius. The observations are so variable that they neither confirm nor contradict such an increase. The model uses a fixed C:N:$O_2$ ratio for all organic matter of 122:16:-172, and Michaelis-Menten kinetics for growth based on inorganic N uptake by phytoplankton (Buitenhuis et al., 2013a, supplementary material Eq. 8, 9). We therefore need a $K_{1/2}$ for growth rather than for uptake to be consistent with the fixed C:N ratio (Morel, 1987). The available uptake rate data do not include the supporting data to allow conversion to the $K_{1/2}$ for growth. We are only aware of measurements of the $K_{1/2}$ for growth by Stawiarski (2014). Based on the latter values of 0.09 $\pm$ 0.15 $\mu$mol L$^{-1}$ for picoeukaryotes, the $K_{1/2}$ of phytoplankton for $NH_4^+$ was set to 0.1 to 5 $\mu$mol L$^{-1}$, increasing linearly with nominal size (Buitenhuis et al., 2013b). Due to the highly dynamic nature of $NH_4^+$ turnover, the model produces a much smoother distribution of $NH_4^+$ concentrations than the observations, but the large scale pattern of surface $NH_4^+$ concentration shows an increase with latitude, consistent with the observations (Fig. 6), which translates into a cost function of 3.0.

## 2.6 N₂O production

N$_2$O production is implemented as two distinct submodels. The diagnostic submodel is based on statistical relationships of $\Delta$N$_2$O/AOU ratios taken from observations and has previously been published (Suntharalingam et al., 2000, 2012). In oxic waters it uses one ratio to estimate the open ocean source of N$_2$O production. In hypoxic waters it uses a higher ratio to represent the increased yield of N$_2$O from both nitrification and denitrification in oxygen minimum zones. The hypoxic N$_2$O yield is maximal at 1 $\mu$mol O$_2$ L$^{-1}$, and decreases with an e-folding concentration of 10 $\mu$mol O$_2$ L$^{-1}$ (Suntharalingam et al., 2000, 2012, supplementary material Eq. 69, 35, 67). Previous studies using regional databases have found different oxic ratios (Suntharalingam and Sarmiento, 2000, and references therein). Therefore, both the oxic and hypoxic ratios have been reoptimised to the global databases (Sect. 3.1 - 3.2).

The prognostic submodel presented here is based on process understanding and explicitly represents the primary N$_2$O formation and consumption pathways associated with the marine nitrogen cycle (Fig. 1). It includes the production of N$_2$O during oxic nitrification (blue arrows in Fig. 1) and during hypoxic denitrification (red arrow in Fig. 1); and a consumption term during denitrification at even lower (suboxic) O$_2$ concentrations (yellow arrow in Fig. 1). The ratios of the three processes are globally invariant (supplementary material Eq. 70, 61, 63, 71). The functional form of the O$_2$ dependence of N$_2$O consumption (suppl. Eq. 71) was the same as that of denitrification (suppl. Eq. 67), and with an O$_2$ response function that is 1.5 $\mu$mol L$^{-1}$ lower than that of denitrification, which is similar to that used by Babbin et al. (2015). We indenpendently optimised the ratios of N2O production and consumption from denitrification (Sect. 3.1), which controls the net N2O production as a function of O$_2$ concentration. There is not enough information at present to optimise the O$_2$ concentration parameters of denitrification and N$_2$O consumption as well. The low O$_2$ ratios of both submodels (supplementary material Section 8.7) were optimised using the database of observed N$_2$O concentration (Sect. 3.1) and the oxic ratios of both submodels were optimised using the database of observed $\Delta$pN$_2$O (Sect. 3.2). The N2O concentrations from both the diagnostic and the prognostic submodels are transported in the same way by physical transport and the formulation of their gas exchange is also identical.

## 2.7 N₂O flux and simulation setup

N$_2$O is transported like other tracers. N$_2$O flux (=air-sea gas exchange) is calculated as:

$$N_2Oflux = (pN_2O_{atm}*K0*(1-p_{watervapor})-pN_2O)*piston\_velocity*\sqrt{660/Schmidt\_number_{N_2O}}*(1-ice\_cover)$$

$$(4)$$

, in which K0 is the solubility (Weiss and Price, 1980), p$_{watervapor}$ is the water vapor pressure (Sarmiento et al., 1992), piston velocity = 0.27*(wind speed)$^2$ (Sweeney et al., 2007), which is optimised for use with the NCEP reanalysis data used here, the Schmidt number for N$_2$O was taken from Wanninkhof (1992), and the ice cover is calculated by the sea ice model LIM2.

In most of the simulations, atmospheric pN$_2$O was calculated from Eq. 2. For the optimised low O$_2$ production we also ran a series of simulations with the NOAA pN$_2$O$_{atm}$ observational data that included seasonal and latitudinal variations (see Sect. 2.2 for the ftp address where we downloaded the data, and Sect. 3.2 for the results). Between 2000 and 2014, we used the

monthly observations for the 12 available latitudes. Monthly anomalies relative to the global average were calculated at each available latitude from the 2000-2016 observations. These were added to Eq. 2 from 1965 and 1976, and to the global average observations between 1977 and 1999. In the model simulation, the data were linearly interpolated between the 12 latitudes and monthly observations.

The PlankTOM10.2 biogeochemical model coupled with the two $N_2O$ submodels is incorporated into the ocean general circulation model NEMO v3.1 (Madec, 2008). The model resolution is $2°$ in longitude, on average $1.1°$ in latitude and has 30 vertical layers, from 10 m in the top 100 m to 500 m at 5000 m. The model simulations were initialised in 1965 from observations (Le Quere et al., 2016b), with $NH_4^+$ initialised as 0, and $N_2O$ initialised from a horizontal interpolation of the MEMENTO observations (see Sect. 2.2). Simulations were run to 2014, forced with daily atmospheric conditions from the
NCEP reanalysis (Kalnay et al., 1996), (for details see Buitenhuis et al., 2013a). Results are reported averaged over the last 5 years.

## 2.8 Estimation of global $N_2O$ flux from point measurements of $\Delta pN_2O$

In previous versions of the PlankTOM model (Buitenhuis et al., 2006, 2010, 2013a) we have used Eq. 3 to evaluate the model because it minimises relative error, which we have found to be more appropriate when the observations span several orders of
magnitude. Unfortunately, statistical confidence intervals have only been defined for $\chi^2$-statistics such as Eq. 5 and 6, which minimise absolute error, so that we end up with 2 cost functions (Eq. 3, 5), depending on the application. To estimate the global air-sea flux of $N_2O$ that best fits the $\Delta pN_2O$ data, and its $\pm1$-sigma (68%) confidence interval, we use the formula described in Buitenhuis et al. (2013a):

$$MSE/MSE_{min} = 0.468 \times n/(n-2) \times \sqrt{(2(2n-2)/(n(n-4))) + n/(n-2)} \tag{5}$$

, in which MSE is mean square error:

$$MSE = \frac{\Sigma(model(longitude, latitude, month) - observation(longitude, latitude, month))^2}{n} \tag{6}$$

, $MSE_{min}$ is the MSE of the model simulation that is closest to the observations, and n is the number of gridded observations.

  In addition to the uncertainty that arises from the model-observations mismatch, uncertainty is contributed by the uncertainties in the $N_2O$ solubility and the piston velocity, the two quantities that connect the measured $\Delta pN_2O$ to the estimated air-sea
flux. The uncertainty in the solubility has been estimated as 3% (Cohen and Gordon, 1978). The uncertainty in the piston velocity has been estimated at 32% (Sweeney et al., 2007). Uncertainties in the solubility and piston velocity are proportional to uncertainty in the optimised $N_2O$ air-sea exchange because the optimised $N_2O$ production needs to change proportionally with solubility and piston velocity to achieve the same $\Delta pN_2O$.

## 3 Results

### 3.1 N₂O production at low O₂

The global $N_2O$ production rate in oxygen minimum zones (OMZs) was optimised using the depth-resolved $N_2O$ data of the MEMENTO database. As noted in previous model studies of ocean $O_2$, global models do not well represent the extent and intensity of OMZ regions (Bopp et al., 2013; Cocco et al., 2013). The modeled OMZs in PlankTOM10 occur at greater depths than observed, resulting in unrealistic vertical distributions of $N_2O$ (results not shown). Therefore, following Suntharalingam et al. (2012), the model was run using fixed observed $O_2$ concentrations (Bianchi et al., 2012), which corrected, in part, the vertical distribution of $N_2O$ production from the two submodels, though it still occurred at too great depths (Fig. 7). In the equatorial regions and in the Pacific ocean the $N_2O$ concentrations are underestimated between ~200 and ~1500 m. depth, and overestimated below that. This shortcoming is not significantly improved in the prognostic model (Fig. 7), even though the prognostic model is more detailed, separately representing the processes of $N_2O$ production and consumption at low $O_2$ concentrations. The depth of maximum $N_2O$ in the model is generally deeper than observed, suggesting that organic matter remineralisation may be too low at shallow depths. This is confirmed by the depth profile of $NO_3^-$, which is underestimated relative to the WOA2009 observations between 100 and 1500 m., and overestimated at greater depths (Fig. 8). In both submodels, the $N_2O$ concentrations in the deep sea are also too high, but since only 5% of $N_2O$ production occurs below 1600 m this does not have a big impact on the global $N_2O$ fluxes. The addition of $N_2O$ consumption in the prognostic $N_2O$ model does result in improvement of the $N_2O$ depth profiles in the Indian Ocean.

In order to find the optimal $N_2O$ production that minimizes the MSE (Eq. 5), we ran a range of simulations in which the low $O_2$ $N_2O$ production was varied in the diagnostic model (Fig. 9A), and a range of simulations in which both the hypoxic $N_2O$ production and the suboxic $N_2O$ consumption were varied in the prognostic model (Fig. 9B). The optimum solution for the prognostic model was found at a gross production of 0.33 Tg N y$^{-1}$. The optimised (net) $N_2O$ production in low $O_2$ regions and its confidence interval were $0.16 \pm 0.13$ Tg N y$^{-1}$ for the diagnostic model, and $0.12 \pm 0.07$ Tg N y$^{-1}$ for the prognostic model. In the optimised diagnostic model the hypoxic $N_2O$ ratio (i.e. net production) is 1.7 mmol $N_2O$ (mol $O_2$)$^{-1}$. In the optimised prognostic model the maximum $N_2O$ production ratio (i.e. gross production from hypoxic denitrification) is 15.4 mmol $N_2O$ (mol $NO_3^-$)$^{-1}$ decreasing to 0 above 34 $\mu$mol $O_2$ L$^{-1}$. The maximum $N_2O$ consumption ratio (from suboxic denitrification) is 15 mmol $N_2O$ (mol $NO_3^-$)$^{-1}$, decreasing to 0 above 28 $\mu$mol $O_2$ L$^{-1}$. This leads to net production that is always positive and has a maximal ratio of 183 $\mu$mol $N_2O$ (mol $NO_3^-$)$^{-1}$ at 10 $\mu$mol $O_2$ L$^{-1}$.

### 3.2 N₂O flux

We used the surface $\Delta pN_2O$ distribution to constrain the total global $N_2O$ flux, and the uncertainty arising from the model-data mismatch (the uncertainties arising from solubility and piston velocity are added at the end). $\Delta pN_2O$ provided a better constraint than the $N_2O$ concentration distribution, since more $N_2O$ production mostly leads to more $N_2O$ outgassing to the atmosphere rather than a significant increase in shallow $N_2O$ concentrations (data not shown). This is because outgassing is proportional to $\Delta pN_2O$, but $N_2O$ concentration is proportional to $pN_2O$, and $\Delta pN_2O/pN_2O$ is small in most of the surface

ocean. The zonal average surface $\Delta pN_2O$ distribution was well simulated by both submodels (Fig. 10D), and the model ensemble covered a wide range of global $N_2O$ fluxes (Fig. 11). The total $N_2O$ flux that best reproduced the $\Delta pN_2O$ distribution was $2.4 \pm 0.3$ Tg N y$^{-1}$ for the diagnostic sub-model and $2.5 \pm 0.3$ Tg N y$^{-1}$ for the prognostic sub-model (Fig. 11). In the diagnostic model, the optimised oxic $\Delta N_2O/AOU$ ratio was 10.6 $\mu$mol $N_2O$ (mol $O_2$)$^{-1}$. In the prognostic model, the optimised oxic nitrification ratio was 123 $\mu$mol $N_2O$ (mol $NH_4^+$)$^{-1}$. The results were the same in both diagnostic and prognostic submodels for the 2000-2004 and 2005-2009 averages, showing that the model was sufficiently spun up.

High $N_2O$ fluxes have been reported for the coastal ocean (Bange et al., 1996), and near-shore upwelling regions (e.g. Arevalo-Martinez et al., 2015). To test whether these regions contribute more to the global $N_2O$ flux than their surface area would suggest, we did the optimisation separately for the coastal ocean ($\leq$200 m bottom depth) for the near-shore non-coastal ocean ($\leq 2°$ from land, >200m bottom depth) for the East Tropical Pacific ($180°$ - $70°$W, $5°$S - $5°$N, >$2°$ from land), and the rest of the open ocean (Table 2). The results show that the coastal ocean contributes only 2% of the global $N_2O$ flux, less than would be expected from its surface area, although there are also fewer observations in the coast (2% of the total) so that the relative error is slightly higher. The near-shore non-coastal ocean contributes 14% of the global $N_2O$ flux both submodels, hardly more than its areal percentage (13%), and it's also fairly well sampled (12% of the observations). The East Equatorial Pacific ocean contributes 27% in the diagnostic submodel and 25% in the prognostic model, more than its areal percentage (22%), and it's undersampled (17%). The open ocean contributes 57 - 59%, slightly less than its areal percentage (61%). This is as expected, because we've separated out the main $N_2O$ hotspots, but the differences are quite small.

When we used observed atmospheric $pN_2O$ that varied with latitude and month (see Sect. 2.2) the results were essentially the same, with an $N_2O$ flux of $2.4 \pm 0.3$ Tg N y$^{-1}$ for the diagnostic sub-model and $2.6 \pm 0.3$ Tg N y$^{-1}$ for the prognostic sub-model (data not shown).

Finally, we add the uncertainties in the solubility and the piston velocity to the total $N_2O$ flux through error propagation. This gives a total uncertainty of $2.4 \pm 0.8$ Tg N y$^{-1}$ for the diagnostic sub-model and $2.5 \pm 0.8$ Tg N y$^{-1}$ for the prognostic sub-model.

## 4 Discussion

Cohen and Gordon (1979) estimated global $N_2O$ production directly from N cycle observations. However, they did not have information on the f-ratio, so their estimate was based on total N assimilation in primary production. We use an updated estimate of primary production and it's error Buitenhuis et al. (2013a), and compile a database of the f-ratio (Fig. 2). We also use a much larger database of the $\Delta N_2O/AOU$ ratio (Fig. 3). We recalculate the N-cycle-based $N_2O$ production based on these extended databases. We find that we can estimate all the relevant steps in the N cycle with observational data, including their uncertainty (Sect. 2.1). At present this uncertainty is still fairly large, at $4.6 \pm 3.1$ Tg N y$^{-1}$. The uncertainty in this estimate is similar to that in Cohen and Gordon (1979), but our uncertainty is based on the uncertainty in all components of the calculation, while their uncertainty was based only on the uncertainty in the $\Delta N_2O/AOU$ ratio. The upper 60% of our estimate overlaps with the lower 62% of the Cohen and Gordon (1979) estimate. The biggest contributor to our uncertainty

is the f-ratio, especially in the tropics, which constitute 44% of the ocean surface area, and additional measurements and/or data-synthesis could help constrain the $N_2O$ budget. The f-ratio data is only based on uptake of $NO_3^-$, $NH_4^+$ and urea, whereas phytoplankton can also take up $NO_2^-$ and organic N (other than urea). One of the major sources of uncertainty in using the $\Delta N_2O/AOU$ ratio is that it is conceptually based on the $N_2O$ production during nitrification, which uses $O_2$. $N_2O$ production

during denitrification is spatially separated from the associated $O_2$ use that is needed to nitrify the $NO_3^-$, the electron acceptor in denitrification. This $NO_3^-$ is produced by nitrification, so in terms of mass balance our calculation is still valid, but this $N_2O$ production would show up as a vertical increase in $N_2O$ without associated increase in AOU at low $O_2$ concentrations (high AOU) in Figure 4. This estimate of global marine $N_2O$ production derived from analyzing the N cycle ($4.6 \pm 3.1$ Tg N y$^{-1}$) is statistically indistinguishable from the $N_2O$ flux derived from $\Delta pN_2O$ observations (2.4 - $2.5 \pm 0.8$ Tg N y$^{-1}$), but has a much

larger error. However, further observational constraints could not only reduce the error, but also extend our understanding of the whole N cycle, including the option of evaluating the model representation of these N cycle processes against observations, and not just the part that $N_2O$ plays in them. Such further constraints are also likely to provide the most productive way to reduce unexplained variability that is found in the observations but not in the present models. E.g., we have shown that both the $N_2O$ and $NO_3$ are underestimated at ~300 - 1500 m depth and overestimated below ~2000 m (Fig. 6, 7). Thus, improved

representation of mesopelagic remineralisation might lead in improved representation of the $N_2O$ depth distribution. However, this falls outside the scope of this study.

Models of the global marine C cycle have been in use for decades, and a lot of the available information has been synthesized, cross-correlated and interpreted in detail (Le Quere et al., 2016a; Buitenhuis et al., 2013b). While actual measurements of N utilisation and transformation have also been made in abundance (Fig. 2, 3, 4, 5A, 6, 7, 9A), the synthesis and global modelling

of these data is less advanced. In addition, N occurs in many different oxidation states in the marine environment (e.g. organic matter and $NH_4^+$ as -3, $N_2$ as 0, $N_2O$ as 0 and +2, $NO_2^-$ as +3, and $NO_3^-$ as +5). Therefore, redox reactions complicate the representation of the N cycle a good deal. This lack of data synthesis and of identification of the most important controls in a complex system is reflected in a relatively low ability of the model to model observed nitrification rates and to a lesser extent $NH_4^+$ concentrations (Table 1).

This lack of knowledge also means that partitioning the global marine $N_2O$ production over the nitrification and denitrification pathways is poorly constrained. Both the diagnostic and the prognostic models assign a small percentage of the total $N_2O$ production to the denitrification pathway, 6 and 4% respectively. However, because of the large bias between the observed and modeled $N_2O$ concentration depth profiles (Fig. 7) these may be underestimates (Suntharalingam et al., 2012; Arevalo-Martinez et al., 2015). Possibly because of the model bias (Fig. 7, 8), the addition of $N_2O$ consumption in the prognostic

submodel does not lead to a significantly better distribution of $N_2O$ across depth or between different basins (Fig. 8). As a result, the $\Delta pN_2O$ distributions are also quite similar (Fig. 10, 12) and the optimised $N_2O$ flux and confidence intervals of the two submodels are also quite similar (Fig. 11). However, it should also be noted, first, that the optimization using surface $\Delta pN_2O$ agrees with the optimization using $N_2O$ concentration that the contribution of the low $O_2$ $N_2O$ production needs to be low (Fig. 11). Second, the error contribution from the model vs. observed $\Delta pN_2O$ comparison is low, with confidence intervals

of 0.3 Tg N y$^{-1}$ for both submodels. Third, $\Delta pN_2O$ is equally well modelled above the low $O_2$ regions as in the rest of the

ocean (Fig. 10, 12), and the contribution of the coastal and near-shore non-coastal ocean are nearly proportional to their surface areas (Table 2). These three features are supporting evidence for our results that suggest that the low $O_2$ regions make a small contribution to the global ocean $N_2O$ production. They should be balanced against the model bias of the vertical distribution of $N_2O$ concentrations, which suggests a larger contribution from the low $O_2$ regions. Freing et al. (2012) also estimated a small fraction of 7% of the global total contributed by denitrification / low $O_2$ $N_2O$ production. Two complementary approaches could provide better constraints: a better representation of the vertical distribution of export and remineralisation would allow the optimization against $N_2O$ concentration observations to achieve better results. But conversely, with better constraints on the physiology of nitrifiers and denitrifiers the $N_2O$ concentration database could provide constraints on the representation of remineralisation. Although there are relatively few $N_2O$ concentration observations, nitrification and denitrification respond to specific environmental queues (in particular $O_2$ concentration), so that the they could contribute a relatively large observational constraint over the full range of environmental conditions.

Despite these shortcomings, the global marine $N_2O$ flux is well constrained to 2.4 - 2.5 $\pm$ 0.8 Tg N y$^{-1}$ by both submodels (Fig. 11). This constraint reflects the fact that the integrated effect of the different physical and biogeochemical processes determines the surface $\Delta pN_2O$ distribution (Fig. 10), so that the integrated total can be well constrained even if the individual processes are not. The $N_2O$ flux is at the lower end of previous estimates, and with a similar confidence interval to other recent estimates (Fig. 4). The confidence interval is dominated by uncertainty in the piston velocity (32%) rather than model-observation mismatches (12%). Because of differences in methodology it is not possible to provide reasons for why our estimate is lower than the more recent estimates. We can, however, compare our estimate to that of (Nevison et al., 1995), because it is also based on a database of $\Delta pN_2O$. Compared to their high end estimate using the piston velocity of Wanninkhof of 5.2 $\pm$ 3.6 Tg N y$^{-1}$, our estimate is lower because we use the more recent 13% lower estimate of piston velocity of (Sweeney et al., 2007), and because our $\Delta pN_2O$ of 7.6 $\pm$ 18.1 ppb is 25 - 28% lower compared to 10.55 natm in Nevison et al. (1995) (the range is calculated based on the water vapor correction for conversion between ppb and natm, which increases from 0.6 - 4.1% at temperatures from 0 - 30 °C, which brings the values slightly closer together).

We also tested how much influence sampling biases of very high supersaturation values might have on the estimated air-sea exchange. If the 40 $\Delta pN_2O$ measurements in the gridded database that are higher than 100 ppb (Fig. 12) are doubled, the optimised $N_2O$ air-sea exchange becomes 2.8 $\pm$ 0.5 Tg N y$^{-1}$ for the diagnostic model and 3.1 $\pm$ 0.5 Tg N y$^{-1}$ for the prognostic model. If the 24 $\Delta pN_2O$ measurements in the gridded database that are higher than 152 ppm are excluded, to decrease the frequency of the highly oversaturated observations down to what both submodels simulate (Fig. 12), the optimised $N_2O$ flux become 2.0 $\pm$ 0.2 for the diagnostic model and 2.3 $\pm$ 0.2 Tg N y$^{-1}$ for the prognostic model. These results still fall within the confidence intervals of the results using the complete database.

Possible biases in ocean physical transport could in theory affect $N_2O$ production in low $O_2$ regions. The indirect impact of ocean physics on low $N_2O$ production through its impact on the distribution of $O_2$, which Zamora and Oschlies (2014) have shown to be substantial, is not quantified here because we used observed $O_2$ (Bianchi et al., 2012) instead of modeled $O_2$. Our model results suggest that the model representation of ocean physics is adequate for the purpose of estimating $N_2O$ flux from biogeochemical model perturbations. On the one hand, if the model had too much ventilation in the OMZs, shallow $N_2O$

concentrations would be underestimated, as they are in the model (Fig. 7), but this would also lead to $\Delta pN_2O$ overestimation in the surface areas above the OMZs, which is not the case. The high $\Delta pN_2O$ are generally lower but spread over a larger area than in the observations (Fig. 10), with a good frequency distribution of high $\Delta pN_2O$ (Fig. 12). On the other hand, if the model had too little ventilation in the OMZs, the optimization would reduce $N_2O$ production in the OMZs in compensation, but the

optimization to $\Delta pN_2O$ would then estimate a higher OMZ $N_2O$ production than the optimization to the $N_2O$ depth profiles to compensate for the low transport, and this is also not the case. Therefore we conclude that potential biases in ocean physical transport do not appear to have a large direct impact on low $N_2O$ production.

Possible biases in ocean physical transport could in theory affect $N_2O$ production in low $O_2$ regions. However the model results do not suggest strong biases in $N_2O$ production as a result. On the one hand, if the model had too much ventilation in

the OMZs, shallow $N_2O$ concentrations would be underestimated, as they are in the model (Fig. 7), but this would also lead to $\Delta pN_2O$ overestimation in the surface areas above the OMZs, which is not the case. The high $\Delta pN_2O$ are generally lower but spread over a larger area than in the observations (Fig. 10), with a good frequency distribution of high $\Delta pN_2O$ (Fig. 12). On the other hand, if the model had too little ventilation in the OMZs, the optimization would reduce $N_2O$ production in the OMZs in compensation, but the optimization to $\Delta pN_2O$ would then estimate a higher OMZ $N_2O$ production than the optimization to

the $N_2O$ depth profiles to compensate for the low transport, and this is also not the case. Therefore we conclude that potential biases in ocean physical transport do not appear to have a large direct impact on low $N_2O$ production. The indirect impact of ocean physics on low $N_2O$ production through its impact on the distribution of $O_2$, which Zamora and Oschlies (2014) have shown to be substantial, is not quantified here because we used observed $O_2$ (Bianchi et al., 2012) instead of modeled $O_2$.

Global oceanic $N_2O$ emissions estimated using atmospheric inversion methods based on atmospheric $N_2O$ concentrations

tend to be higher than our results (Fig. 4). However, $N_2O$ emissions from inversions in the Southern Ocean are lower than the priors (Hirsch et al., 2006; Huang et al., 2008; Thompson et al., 2014; Saikawa et al., 2014). These low Southern Ocean emissions (0.02 - 0.72 Tg N y$^{-1}$) are consistent with our results (0.68 - 0.79 Tg N y$^{-1}$). South of 30°S, 88% of the Earth surface is ocean, resulting in a clearer attribution in the inversions of the atmospheric $N_2O$ anomalies to ocean fluxes. We suggest that the higher emissions estimates from inversions for the global ocean could be due to a combination of overestimated priors of

ocean fluxes in combination with insufficient observational constraints at latitudes North of 30°S to allow correct partitioning between land and ocean fluxes. Results presented here are for the open and coastal ocean. The largest coastal seas are resolved in our model, although specific coastal processes, such as the interactions with sediments and tides, are not. Our results do not include emissions from estuaries. Fluxes from these could be as large as 2.3 - 3.6 Tg N y$^{-1}$ according to one estimate (Bange et al., 1996), and could be another contributing factor to the difference between our results and those of atmospheric inversions.

*Code and data availability.* The four databases presented in this manuscript are available as NetCDF files from https://www.uea.ac.uk/green-ocean/data. The code of PlankTOM10.2 is available at greenocean-data.uea.ac.uk/model/PlankTOM10.2.tar

*Competing interests.* The authors declare they have no competing interests.

*Acknowledgements.* This research was supported by the European Commission's Horizon 2020 programme through the CRESCENDO and EMBRACE projects (projects 641816 and 282672). We thank Martin Johnson for the database of $NH_4^+$, and Andrew Yool for the database of nitrification rates. The MEMENTO database is administered by the Kiel Data Management Team at GEOMAR Helmholtz Centre for Ocean

5 Research and supported by the German BMBF project SOPRAN (Surface Ocean Processes in the Anthropocene, http://sopran.pangaea.de). We thank Alina Freing for providing the corrected numbers for the polynomial fit to the atmospheric pN2O data, and NOAA for providing atmospheric pN2O data.

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

**Table 1.** Cost function (Eq. 3) for the optimisation simulations of sections 2.2-2.4, relative to the respective observational databases. The nitrification rate in bold was used in this study.

| Database | Model change | Cost function |
|---|---|---|
| Nitrification rate | $0.2 \, \text{d}^{-1}$ | 4.22 |
| | $0.1 \, \text{d}^{-1} \times 2^{(T/10)}$ | 4.18 |
| | $\mathbf{0.79 \, \text{d}^{-1} \times 2^{(T/10)} \times (1 - 0.159 \times \ln(O_2))}$ | **4.16** |
| | $0.58 \, \text{d}^{-1} \times 2^{(T/10)} \times e^{(-0.14 \times I)}$ | 7.15 |
| | $4.7 \, \text{d}^{-1} \times 2^{(T/10)} \times (1 - 0.159 \times \ln(O_2)) \times e^{(-0.14 \times I)}$ | 6.87 |
| Surface $NH_4^+$ concentration | $K_{1/2}$ estimated from observations | 3.0 |

**Table 2.** Contributions of coastal (bottom depth $\leq 200$ m), near-shore non-coastal ($\leq 2°$ from land, bottom depth > 200 m), East equatorial Pacific ($180°$ - $70°$W $5°$S - $5°$N, >$2°$ from land) and rest of the open ocean (>$2°$ from land, bottom depth > 200 m, excluding East Eq. Pac.) to $N_2O$ flux, area and number of observations.

| Region | Submodel | $N_2O$ flux | % $N_2O$ flux | % area | % $n_{obs}$ |
|---|---|---|---|---|---|
| Coastal ocean | Diagnostic | $0.05 \pm 0.01$ | 2 | 5 | 2 |
| | Prognostic | $0.041 \pm 0.007$ | 2 | | |
| Deep offshore | Diagnostic | $0.33 \pm 0.04$ | 14 | 13 | 12 |
| | Prognostic | $0.37 \pm 0.04$ | 14 | | |
| East Eq. Pac. | Diagnostic | $0.64 \pm 0.05$ | 27 | 22 | 17 |
| | Prognostic | $0.67 \pm 0.05$ | 25 | | |
| Open ocean | Diagnostic | $1.37 \pm 0.19$ | 57 | 61 | 69 |
| | Prognostic | $1.54 \pm 0.21$ | 59 | | |

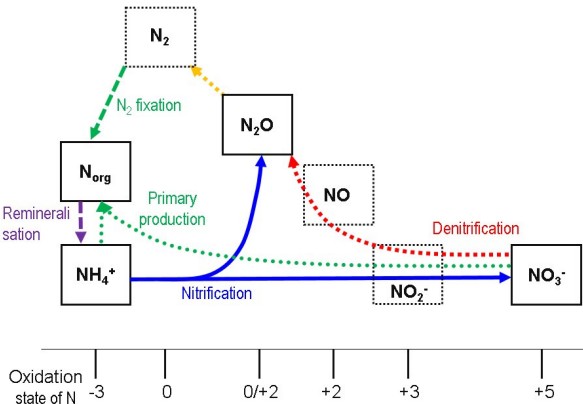

**Figure 1.** Primary biological pathways of the oceanic nitrogen cycle represented in the model simulations, along with redox states of N. Nitrification occurs in the oxic ocean (blue arrow). Denitrification yields net $N_2O$ production in hypoxic conditions (red arrow) and net $N_2O$ consumption in suboxic conditions (yellow arrow). Only organic nitrogen ($N_{org}$), $NH_4^+$, $NO_3^-$ and $N_2O$ are represented as model state variables.

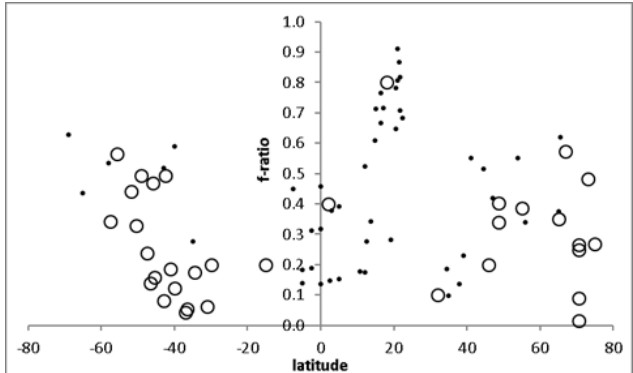

**Figure 2.** f-ratio ($\rho_{NO_3^-}/(\rho_{NO_3^-}+\rho_{NH_4^+}+\rho_{urea})$) as a function of latitude, from [15]N uptake experiments. Small dots were estimated without measuring $NH_4^+$ or urea concentrations (Prakash et al., 2008, 2015; Gandhi et al., 2010, 2012). Large dots did not give a significant linear relationship with absolute value of latitude, and were therefore averaged at $0.29 \pm 0.18$ (Wafar et al., 2004; Varela et al., 2005, 2013; Joubert et al., 2011; Thomalla et al., 2011; Simpson et al., 2013).

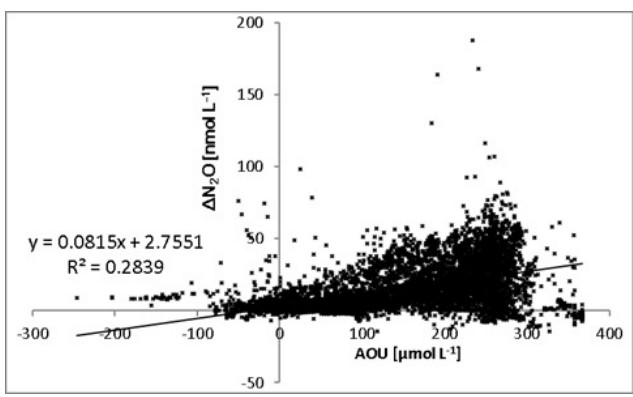

**Figure 3.** Apparent $N_2O$ production ($\Delta N_2O$ nmol $L^{-1}$) as a function of apparent oxygen utilization (AOU $\mu$mol $L^{-1}$).

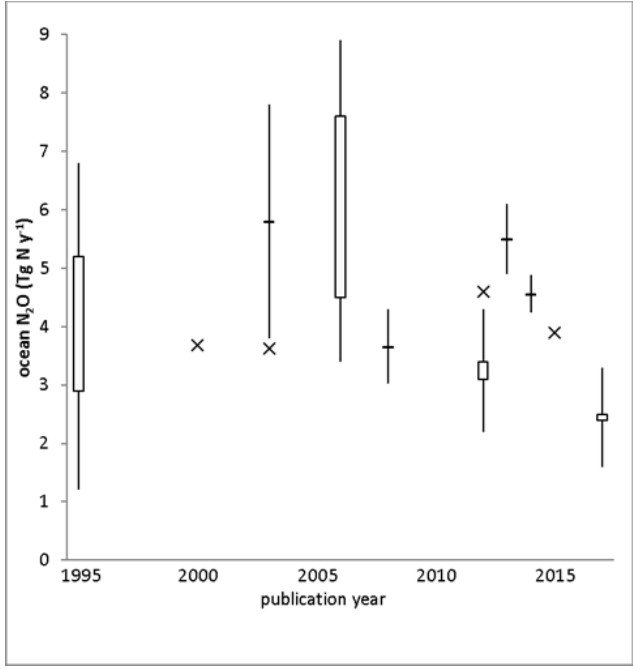

**Figure 4.** Published estimates of global ocean $N_2O$ production or air-sea exchange. Estimates based on global observational datasets shown as boxes when ranges are given and whiskers if error estimates are given (ocean observations: Nevison et al. (1995, 2003); Freing et al. (2012) (plotted in 2011), Bianchi et al. (2012), this study; atmospheric inversions: Hirsch et al. (2006); Huang et al. (2008); Thompson et al. (2014) (plotted in 2013), Saikawa et al. (2014)), model estimates shown as crosses (Suntharalingam and Sarmiento (2000); Jin and Gruber (2003); Suntharalingam et al. (2012); Martinez-Rey et al. (2015)).

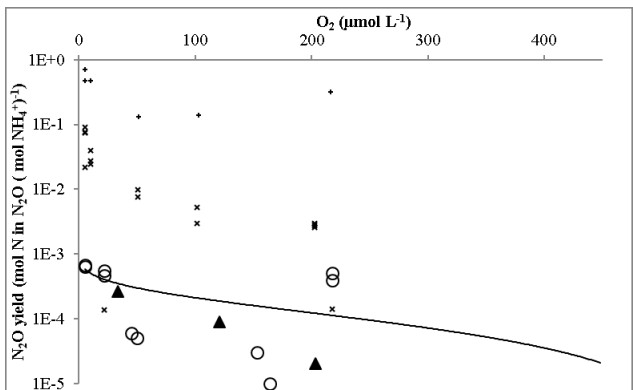

**Figure 5.** $N_2O$ yield of nitrification (N atom:atom) as a function of $O_2$ concentration, filled triangles: AOA (Loescher et al., 2012), open circles: AOB at low to medium cell numbers (Frame and Casciotti, 2010; Loescher et al., 2012), crosses: marine AOB at high cell numbers (Goreau et al., 1980; Frame and Casciotti, 2010), plusses: soil AOB at high cell numbers (Lipschultz et al., 1981). Black line: logarithmic fit to AOA and low to medium cell number AOB (yield = 0.791-0.126·ln($O_2$) mmol N in $N_2O$ (mol $NH_4^+$)$^{-1}$).

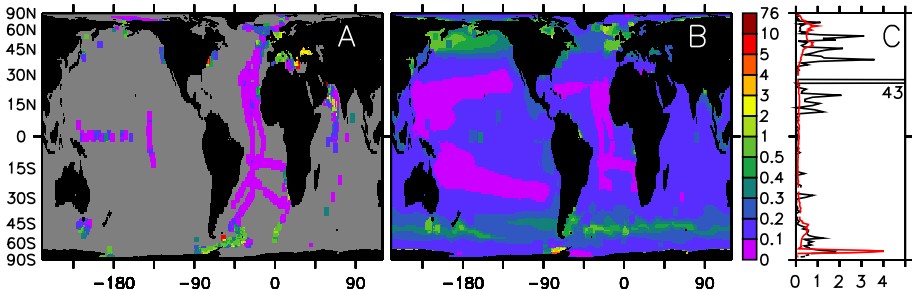

**Figure 6.** Surface $NH_4^+$ concentration ($\mu$mol L$^{-1}$). A) observations (symbol size is $5 \times 5°$). B) model results are for the same months where there are observations, and annual averages everywhere else. C) zonal average, black) observations, red) model results. Model results are for the same months and longitudes as the observations. Latitude y-axis to the left of panel A.

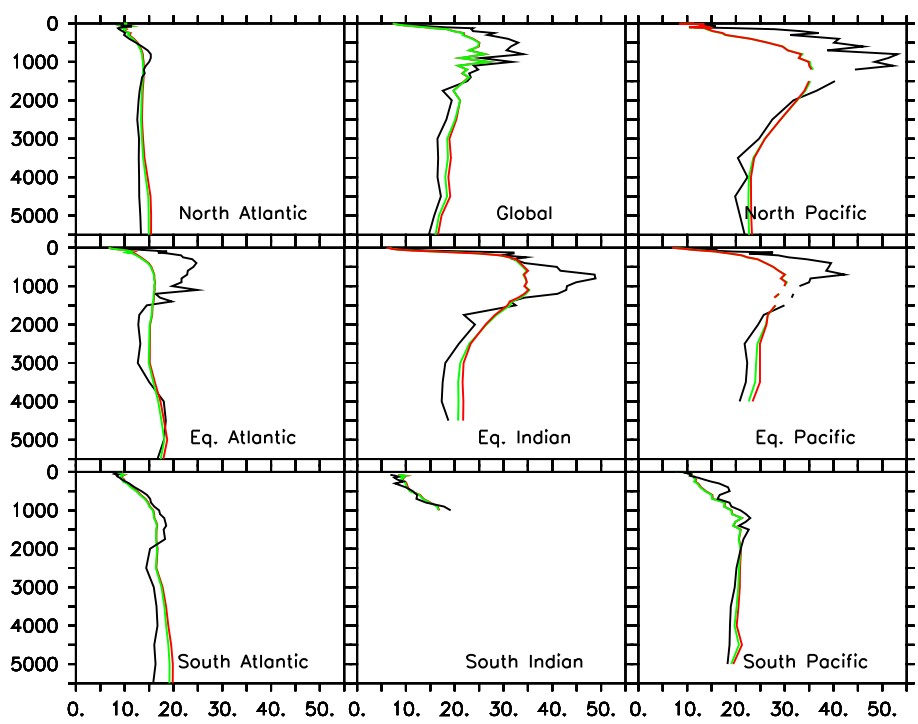

**Figure 7.** Depth profiles of $N_2O$ concentration (nmol $L^{-1}$) for different basins. Black lines: observations, Green lines: optimised diagnostic model, Red lines: optimised prognostic model.

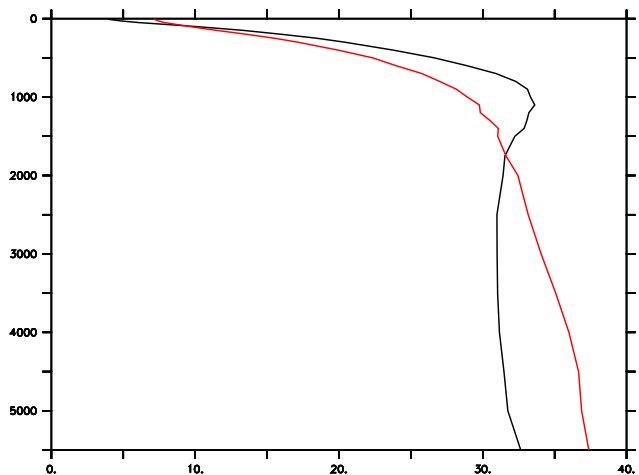

**Figure 8.** Depth (m.) profile of average $NO_3^-$ concentration ($\mu$mol $L^{-1}$). Black line) WOA2009 synthesis of observations, not interpolated. Red line) Model results sampled at the places where there are observations.

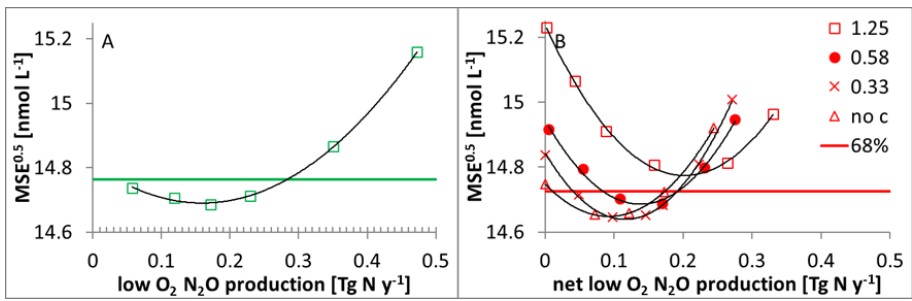

**Figure 9.** $MSE^{0.5}$ for the two $N_2O$ submodels compared to the $N_2O$ concentration database as a function $N_2O$ production in the low $O_2$ regions. $MSE_{min}$ was obtained as the minimum of a second order polynomial fit (black lines). The $1\sigma$ confidence interval, where MSE equals the value calculated from Eq. 5, is indicated by the horizontal lines. A) diagnostic submodel, each point represents a simulation with a different low $O_2$ ratio, B) prognostic model, "no c" is with no $N_2O$ consumption i.e. net production = gross production. All other lines have a constant gross production, and net production varies with different $N_2O$ consumption rates. Range of parameter values is given in the supplementary material Section 8.7.

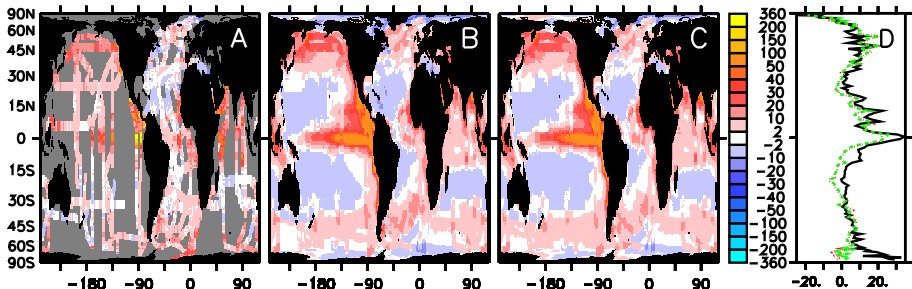

**Figure 10.** Surface $\Delta pN_2O$ (ppb). A) observations (symbol size is $5 \times 5°$), B) optimised diagnostic model, C) optimised prognostic model. Model results are for the same months where there are observations, and annual averages everywhere else. D) zonal average, Black line: observations, Green dashed: diagnostic model, Red dotted: prognostic model. Model results are for the same months and longitudes as the observations. Latitude y-axis to the left of panel A.

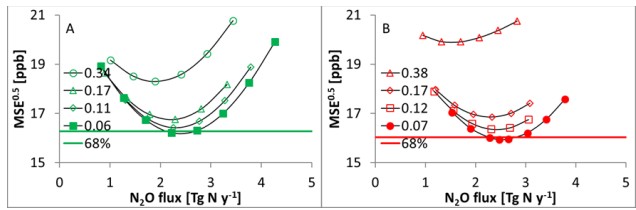

**Figure 11.** $MSE^{0.5}$ for the two $N_2O$ submodels compared to the $\Delta pN_2O$ database as a function of global $N_2O$ flux at different (net) $N_2O$ production rates in the low $O_2$ regions. $MSE_{min}$ and confidence intervals as in Fig. 8. A) diagnostic submodel, the four lines represent the four best low $O_2$ production rates from Fig. 9A, each point represents a simulation, different symbols indicate different low $O_2$ ratios, points with the same symbols have different oxic $N_2O$ production ratios. B) prognostic submodel, the four lines represent the optimised net production rates at the four best gross production rates from Fig 9B, points with the same symbols have different $N_2O$ ratios for nitrification.

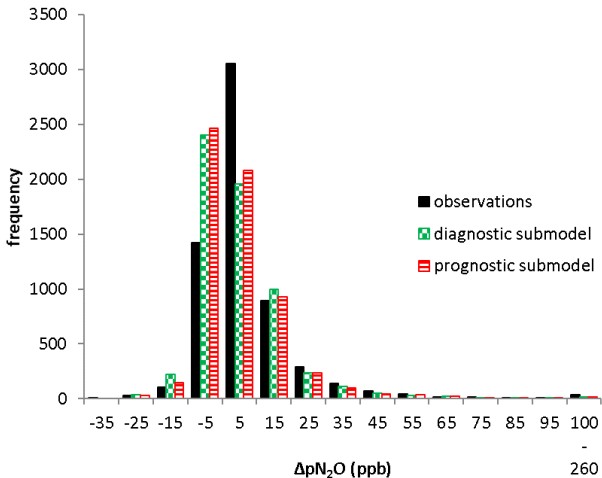

**Figure 12.** Frequency distribution of $\Delta pN_2O$ in the observations (solid black), and the optimised simulations of the diagnostic submodel (green squares) and the prognostic submodel (red lines).