# Peer review of "Constraints on global oceanic emissions of $N_2O$ from observations and models"

_Biogeosciences, 2017_

## Referee Comment (RC1) · Anonymous Referee #1 · 4 Jun 2017

In this manuscript, the authors present newly estimated global ocean $N_2O$ flux to the atmosphere and its confidence interval using observations and two submodels of $N_2O$ production. The paper provides interesting insights but the writing could be improved to make the manuscript clearer. The main problem of the paper, as I see it, is that there are not enough details to assess the validity of the model and results. Below are some major comments and questions, followed by minor edits.

Major comments/questions:

It is unclear how the authors calculate the best estimate of $N_2O$ production using observations (l. 82). How is the range obtained in this case? I thought that the authors might be using the maximums and the minimums of each factor to calculate the range but that does not seem likely.

I am having hard time understanding the equation 1. How is this equation derived and why are such large significant figures used? This equation does not account for the latitudinal dependence of $pN_2O$ - wouldn't that be a problem? Isn't it better to use atmospheric model results validated by atmospheric measurements of $N_2O$?

I think there might be a mistake in equation 2. Otherwise, I do not see how a value of 2 could mean that the model deviates form the observations by a factor of 2 in either direction. $10^{\wedge}(10\log 2) = 1024$ and it is nothing close to a value of 2. Please explain. Perhaps, the standard mathematical notation (summation and the number of observations n rather than "average") would be more appropriate here.

It would be useful if the $N_2O$ flux calculation in section 2.7 is explained in a little more detail, rather than stating that it "is calculated with the piston velocity from Sweeney et al. (2007)." I am not familiar with this calculation and would love more explanations on how the ocean $N_2O$ flux is estimated but the Sweeney et al. (2007) is not listed in the references either.

I am not sure how equation 3 is used to determine the global air-sea flux of $N_2O$ that best fits the $\Delta pN_2O$ data, if $RSS/RSS_{min}$ just depends on the number of observations. I do not understand how different model simulations would have different values of $RSS/RSS_{min}$ if the number of observations is the same.

As for equation 4, I think that its application should be described within the methodology section, rather than just mentioning a little in the discussion section.

Also, how did the authors optimize various model parameters? And is it not a problem that the optimized oxic $\Delta N_2O/AOU$ slope of 12.7 μmol $N_2O$ (mol $O_2$)$^{-1}$ is so different from the global average given earlier in lines 77-78 (81.5 ± 1.4 nmol/mmol)? What is the value for this parameter in the prognostic model?

Minor comments
    1. L. 24 "It also currently" → "It is also currently"

2. There are several places in the text, where more detailed or clearer explanations would help readers understand the paper better. For example, l. 53-56 is unclear what the sentence means. Do the authors mean that $\Delta N_2O/AOU$ slope becomes negative under suboxic conditions and that leads to the ambiguity of how much $N_2O$ is produced in this region? Please clarify.
3. L. 71 "observationally derived" → "observationally-derived"
4. L. 75 Since not all readers of this paper are experts in oceanic biogeochemistry, it would be helpful to explain that the f-ratio is the fraction of total primary production by nitrate.
5. L. 79 What is the "-$O_2$:C ratio"? What is the dash for?
6. L. 233 "N cycle based" → "N cycle-based"
7. L. 242-246 "This estimate…" run-on sentence and needs to be rewritten.
8. L. 263-267 "It should also…" run-on sentence and needs to be rewritten.
9. L. 286 "140 pm" → "140 ppm"
10. L. 290-294 "On the one hand…" run-on sentence and needs to be rewritten.
11. N-cycle data database used in this paper are shown as embargoed in the data source pointed by the authors (https://www.uea.ac.uk/green-ocean/data). Will the data be publicly available?

---

## Referee Comment (RC2) · Anonymous Referee #2 · 8 Jun 2017

I strongly support the goal of this paper, to better constrain the oceanic N2O flux using optimization techniques based on a compilation of datasets of N2O and related N cycle variables, combined with process based models. However, the methodology is difficult to follow and it is hard to believe that all 4 data-based approaches converge to basically the same answer and have the same relatively narrow range of uncertainty, which is governed primarily by uncertainty in piston velocity. There is also no overall sense of what sets this paper apart from earlier efforts, since it seems to be based heavily on what is largely the same delta pN2O data set used before. While I support publication in principle, I think there are many details that should be clarified and explored before this paper is ready for publication.

Specific comments L24 Typo: It (is) also currently estimated as the dominant contribu-

tor . . .

L32 It's worth mentioning (up front) that this wide range is governed in large part by the possibility of very large coastal and estuarine fluxes. Later on line 87 we learn that the dataset resolution used here is 1x1 degree or 1.1 x 2 degree (plankTOM10.2, line 164), i.e., probably not good enough to resolve these coastal areas.

L43 although there are additional pathways, such as . . . (please give brief list).

L72 probably should mention up front that the deltaN2O/AOU data are based on ME-MENTO. Otherwise, it's a bit confusing to know the basis of this calculation.

L76 NPP is. . .estimated at 58 +/- 7 PgC/yr. . . based on what? An ocean model? Satellite data? Even at the lower end of 51 PgC/yr, this is on the high side of satellite-based estimates.

L83, please list relevant references rather than just saying "(see Introduction)".

Line 92-93. Since pN2O is close to equilibrium in much of the ocean, it seems important to consider the quality of these pN2O measurements. For example, surface measurements made with underway systems tend to have much higher precision than analyses based on bottle collection. Was the uncertainty comparable across the ME-MENTO database and if not, were the differences in data quality accounted for in the subsequent calculations?

L125 Ocean models often do a poor job of reproducing observed O2. Suntharalingam 2012 used WOA O2 rather than model O2 for that reason. Presumably, the sensitivity to light, temperature and O2 described here is based on values from plankTOM10.2 (if not, please clarify). How well does plankTOM10.2 reproduce O2 relative to observations? (Note: I saw later that my question was addressed in the Results on lines 184-188. That material belongs up front in the methods description.)

L132 paragraph starting here. This paragraph could be written more clearly, especially the sentence spanning L137-138. What is a variable N quota? Is the model using

Michaelis Menten kinetics? On line 144, a low cost function of 3.3 is better than the cost functions of >4 described for the previous model, correct? Yet, the sentence beginning on L142 with "However" suggests a large uncertainty.

L155-156 What is meant by "The slopes of the three processes" ?

Section 2.2-2.8. General question. Do the 4 databases described in section 2.2 correspond to the specific sections 2.4-2.7? If so, where does section 2.8 fit in? Is Equation 3 an alternative cost function to Equation 2 described in Section 2.3?' The apparent switch from Equation 2 to Equation 3 as the optimization technique is confusing and unclear.

Line 190 and Figure 6. The model substantially underestimates N2O in the most important hotspots of production. Doesn't this mean it will tend to underestimate global N2O production? It seems like this concern is dismissed somewhat casually with hand-waving arguments, e.g., the text starting on line 289.

Line 201-202 Please clarify how these results link together. Line 201 says that both hypoxic production AND CONSUMPTION were optimized. The subsequent results mention GROSS production of 0.33 TgN/yr, then optimized N2O production of 0.12 or 0.16 TgN/yr. Are the latter results net production? Can we infer that about 0.17-0.21TgN/yr is consumed in suboxic zones?

Line 202-203 Total production of 0.12-0.33 TgN y-1 in low O2 is only about 10% on average of total production. This is much lower than the 33% suggested by Suntharalingam et al. 2012. Does this mean that the authors are concluding that the OMZs are only responsible for a small fraction of global N2O production? Please expand on this point and call it out more explicitly in the Discussion.

Line 204 perhaps add a clause clarifying that the .0017 molN2O/molO2 slope is about 20 times the mean deltaN2O/AOU ratio of 8.15e-5 from line 82.

Line 205 production for the prognostic model is given in units of umol N2O (mol NO3)-

1. Does this represent NO3- consumed by denitrification, or produced by nitrification? Can you provide an estimate of how this relates to the previous units of mol N2O/mol O2?

Line 216 Please use consistent N2O (mol O2)-1 slope units. Here the units are umol/mol. On line 78 they were nmol/mol. On line 204, they were mol/mol.

Line 217 How does this nitrification slope in units of umolN2O/molNH4+ relate to the "N2O production slope" on line 206 in units of mol N2O/mol NO3-?

Line 219-220 Are all measurements really of deltapN2O, or are most of pN2O in the surface ocean? In the latter case, what is the uncertainty in pN2Oatm, e.g., from Eq 1?

Line 220 On what basis was this 1978 estimate made? Is there updated information that could be used?

Line 233 Typo or confusing sentence.

Line 248-250. It would be good to provide references to support these claims.

Line 282 – paragraph starting here. This exercise, combined with large data gaps in Figure 9a, including in both the ETSP and ETNP, suggest to me that the authors are overstating the degree of certainty in their confidence interval. There are large areas of the ocean with no data, including in the most important hotspots for N2O production.

Line 303-304. This is the first mention of the fact that the model produces low fluxes from the Southern Ocean. Can you cite the relevant figure here and call attention to this point earlier in the Results section? (Figure 9b,c,d all seem to indicate a substantial flux from the Southern Ocean.)

Line 308-310. The neglect of estuaries is indeed a key uncertainty, which needs to be mentioned much earlier, i.e., in the Introduction. It is also debatable whether coastal areas are adequately represented in the models presented here, which 2x1 or 1x1

resolution.

L487 Figure 2. This figure suggests very high f-ratios, e.g., of 0.8-0.9 in the northern subtropical gyres, that are a little hard to believe. The global mean looks to be on the order of 0.4! Are these generally accepted values or are they biased by measurements in highly productive coastal waters?

L505, "Model results are for the same months and longitudes as the observations." What about latitudes?

L527 This plot is dominated by the error bars and somewhat obscures the focus on the mean value, which arguably is the more important quantity. The current study makes a much more detailed effort to quantify uncertainty than most of the previous studies (some of which make no effort at all). Could a separate panel with a narrower Y-axis range be plotted to better compare the mean value of the fluxes? And can you please provide some discussion of the main factors contributing to the differences in mean value?

---

## Short Comment (SC1) · 13 Jul 2017

Thanks for this contribution to global marine N2O modeling. May I ask some questions regarding the model formulation and applied parameter sampling:

Line 148: Is there nitrification at 1 umolO2/l?

Line 156: How is N2O consumption modeled? As a first order consumption term as applied in other studies? How large is gross consumption? What O2 threshold do you use to separate nitrification, production from denitrification and consumption from denitrification? How large are aerobic and anaerobic remineralization fluxes in the model?

Line 166: Are modeled N2O concentrations not drifting substantially after such a spin

up procedure?

Line 199: How many parameter perturbation simulations did you run? Which sampling technique is applied to vary parameters? Over which range are parameters varied? What does the legend in Fig 8/10 stand for? Could you illustrate the sampled slopes and resulting optimal slope? Are fluxes tied stoichiometrically to remineralization fluxes? Why is N2O consumption slope given as N2O/NO3-? Does this make sense stoichiometrically?

Figure 6: Many global N2O modeling studies present N2O versus O2 scatter plots for evaluation. What does this relationship look like in the model?

The N2O flux estimate of 2.4+/-0.8 Tg N yr-1 is much lower than what was reported in Suntharalingam et al. 2000/2012, on which the model builds ('4.6 Tg N yr-1 (comprised of 3.0 Tg N yr-1 from the 'nitrification' pathway, and 1.6 Tg N yr1 from the low-oxygen pathway', Suntharalingam et al. 2012). How come? Does your prior include these previous fluxes? Your N2O production at low O2 is now ~10 times smaller compared to this previous model.

---

## Referee Comment (RC3) · Anonymous Referee #3 · 14 Jul 2017

The manuscript by Buitenhuis and Coauthors describes the results of simulations with an ocean biogeochemical model that includes different parameterizations of N2O production, constrained with available N2O observations. The main finding is a net N2O outgassing to the atmosphere of 2.4 $\pm$ 0.8 TgN/year, which is substantially lower than many of the estimates previously reported, and also less uncertain. A very small proportion of the N2O production comes from denitrification-associated pathways in suboxic waters. The estimate also appears robust to the choice of the parameterization of N2O production.

This is a short and potentially useful paper, although not particularly original. But I think that, if better supported, the results will push other scientists to reconsider estimates of N2O emissions from the ocean (as well as from other sources) in light of the low values suggested. That said, I also think the paper is poorly written, in particular when it comes to the description of the methods employed - for example the model equations, the rational and choices behind the parameterizations, the steps behind the optimization. Furthermore, I have some additional concerns about the results that prevent me from fully supporting publication of the manuscript as is.

Specific comments:

- The model formulation is quite hard to follow, partly because equations are not show. This makes it difficult at times to judge the validity of the model's assumption. I strongly encourage the Authors to show all the pertinent equations, either in the main text, or in an appendix.

- The choice of some of the model equations and parameterization is unclear and could be better justified. The Authors could do a much better job explaining why certain functional forms have been utilized, and what consequences these choices may have, if any. For example, looking at Table 1, line 3 lists an equation that uses the logarithm of $O_2$. Is there any reason for this form? The logarithm will expand the range at very low $O_2$ concentration - do we trust $O_2$ measurements there? Further, this form breaks down at $O_2=0$; does this ever happen in the model/observations, and is there any limit applied to prevent it? Finally, all of these equation should represent a process ultimately limited by $O_2$. Is there any limitation as $O_2$ goes to zero?

- I found the distinction between the prognostic and diagnostic model (for $N_2O$) somewhat confusing. In both models $N_2O$ is carried as a prognostic tracer - except in the first model it does not depend on other N-cycle tracers, and is not consumed, but only produced and passively advected until it outgassed from the surface. What makes one model diagnostic and another prognostic?

- Regarding the prognostic model - the Authors say that it explicitly represent the redox transformations that lead to the conversion of $NH_4^+$ to eventually $N_2O$ (actually only a subset, as for example, the model does not include $NO_2^-$), but the model seems to

still parameterize them heavily. For example the current understanding is that N2O is an obligate intermediary during heterotrophic denitrification, so that one should expect a gross N2O production comparable to the denitrification rate (i.e. ∼70 Tg N/year) However, the Authors indicate a suboxic gross production of only 0.33 Tg N year - a very small rate in comparison. This may be explained by the use of "slopes" in the prognostic model that relates N2O to other tracers (more on these slopes in the next comment). This implicitly assumes a tight coupling between production and consumption at suboxic levels, with only a fraction of the N cycled by denitrification escaping to the water column. It may be fine as a parameterization - especially since it is one that is optimized against observations. However it may be problematic if the model is to be used under varying circulation or climate - the coupling between production and consumption may vary, and given the large gross N2O fluxes this may impact net production and accumulation of N2O.

- It is unclear what "slopes" are used in the prognostic model. Are these slopes actual yields (e.g. N2O production per NH4+ oxidized), relationships with O2 consumed, or just empirical relationships based on data syntheses? And what is then the slope of the third step of N2O cycling (consumption of N2O by denitrification)? Is it a relationship with O2, with NH4+ or with NO3-? (and specifically, is there explicit denitrification in the model, so that one could relate N2O production to NO3- deficit?).

- The lack of spinup in the model is worrisome: the model was apparently initialized in 1965 and run for 49 years through 2014. This is a short running time, and it completely misses a spinup phase. It may very well be the case that the N2O inventory of the ocean over the last 5 years is still adjusting from the initial condition, in a way that could bias the outgassing estimates. For example, there seems to be a substantial accumulation of N2O in the deep ocean - if this is still ongoing after 49 years, then the outgassing estimated by the Authors could be a lower estimate. A comparison between the total net production in the interior and the outgassing could give a sense of any disequilibrium. Note that a similar modeling study by Martinez Rey et al., 2013,

BGS (incidentally finding about 4Tg/year emissions) suggested a 150-year spinup was not enough to eliminate drifts in N2O and other biogeochemical variables. Any drift should be discussed in the paper, and the consequences assessed.

- I found the description of the optimization steps very unclear. It took me a while to figure out what steps the Author follow and how the model is actually compared to the data, and I'm still not sure about them. Now my understanding is that a first optimization is carried out for the NH4-cycle using nitrification rates and NH4+ concentrations; then a second optimization is performed with interior N2O data to determine parameters for low-O2 pathways (but does this apply to both the prognostic and diagnostic model?); and finally a third optimization (presumably with some parameters fixed by the previous steps?) using surface Delta-pN2O data for the global source terms, used to determine the final air-sea fluxes. That's my understanding but I am still not sure I got it right, and some aspects remain puzzling. I think this could be much better explained from the start, for example by a method section outlining the optimization strategy in more detail.

- Related to the previous comment, the equations for the optimization are absolutely opaque and unclear. They need to be substantially clarified: ideally anyone should be able to apply them after reading the paper, which is not the case. For example equation (1) is not very specific: instead of "average", "model", "observations" the actual mathematical form could be given - this would also help knowing how the average was done, wether the in situ or gridded data were used, how the model was sampled etc. Similarly I am completely at loss with section 2.8, and I could not trace back the steps applied by the Authors based on this description alone. How is RSS/RRS_min (equation 3) used, how does it relate to the quantities shown in Fig. 8 and 10, and why does it only contain the number of observations but no information on the actual variables? What does the "phi" term (equation 4) represent, and how is it actually used?

- Regarding the final estimate of N2O air-sea flux, I think it could be couched much better into the context of previous estimates (also, a table would help), and what could be

behind the potential discrepancies in light of the substantially lower revision. This could be especially interesting given that many modeling studies use a similar approach. The Authors also present an "observational" estimate of N2O production whose central value (4.6 TgN/year) is quite different than the final model estimate - this discrepancy could be added to the discussion. I am not particularly surprised by the lack of sensitivity of N2O production to the choice of diagnostic and prognostic models, since both are optimized versus observations. Surface pN2O should be a quite powerful constraint to outgassing fluxes. However, one may still expect different sensitivities to interannual variability and climate change, so this is not a strong argument in favor of not resolving complex pathways that characterize the low-O2 N2O cycle.

- The model is biased in its representation of export and remineralization, as well as N2O distribution. The discussion of the effect of these biases (e.g. lines 289-300) is not especially thorough - so the conclusion, in particular regarding the narrower range of the new estimate, is not very convincing. Furthermore, there are hidden resolution biases. For example, the model can not resolve low-O2 coastal upwelling regions, which have been shown to be powerful conduits to N2O outgassing (e.g. Arevalo Martinez et al., 2015, Nature Geosciences). The abstract/conclusions could be more cautious with respect to the real uncertainties.

- Line 43: The reference to Klawonn et al., 2015 is missing.

- Line 95, equation 2: More information should be given on this equation, and how it was used in the model/observation comparison. Does using this equation mean that the N2O flux is calculated for a specific period, or that it varies in time? This is unclear. Also, there number of significant digits in the various coefficients is way larger than any believable uncertainty associated with the measurements the equations should fit.

- Section 2.4, Table 1. Maybe some effort can be done to evaluate the improvements associated withe each model: by adding terms the cost function decreases minimally - is the improvement significant? Does it justify the increase in the model degrees of

freedom?

- Line 133-134. The equation could be shown.

- Section 2.6. The slopes (of what, with respect to what?) and relationships used for the model should be clarified with equations, and maybe with corresponding figures (e.g. the observational constrains used). Also, what is the range from which the various slopes were drawn in order to run the different model versions for the optimization? How were they determined? What values were actually used? Finally, there must be concentration thresholds associated to the transitions between different slopes (e.g. O2). How were these thresholds determined? Were they also optimized for?

- Line 211-212. The reasoning is unclear: an increase in outgassing for a given atmospheric concentration should be driven by a parallel increase in surface concentrations, since the flux is proportional to the concentration (or pN2O) difference. For example, in the limit of removing the saturation N2O concentration, a doubling of the interior production of N2O should double both the outgassing and the surface concentration.

- Lines 242-247. This entire paragraph is very unclear, please clarify.

- Lines 270-271. Constraining remineralization backwards from N2O production seems a bit far-fetched, given how hard it is to even constrain processes like denitrification alone.

- Lines 279-281. Please clarify.

- Lines 294-297. The issue of biases in model circulation could be assessed by using ventilation tracers, e.g. CFCs. Are they available for this mode?

- Line 308: do the Authors really think their model can capture costal N2O dynamic, and the massive air-sea fluxes observed there (see Arevalo Martinez et al., 2015), especially in eastern boundary upwellings?

---

## Author Comment (AC1) · 25 Oct 2017

In this manuscript, the authors present newly estimated global ocean $N_2O$ flux to the atmosphere and its confidence interval using observations and two submodels of $N_2O$ production. The paper provides interesting insights but the writing could be improved to make the manuscript clearer. The main problem of the paper, as I see it, is that there are not enough details to assess the validity of the model and results.

We thank the reviewer for the comments. We have tried to clarify the methodology throughout the manuscript.

Below are some major comments and questions, followed by minor edits.

Major comments/questions:

It is unclear how the authors calculate the best estimate of $N_2O$ production using observations (l. 82). How is the range obtained in this case? I thought that the authors might be using the maximums and the minimums of each factor to calculate the range but that does not seem likely.

Errors were calculated with standard error propagation; we added the line: "Here and in the rest of the paper, errors were propagated in the usual way:

error = $(((error\ of\ A)/A)^2 + ((error\ of\ B)/B)^2 + \ldots)^{0.5} \times A \times B \times \ldots$"

I am having hard time understanding the equation 1. How is this equation derived and why are such large significant figures used? This equation does not account for the latitudinal dependence of $pN_2O$ - wouldn't that be a problem? Isn't it better to use atmospheric model results validated by atmospheric measurements of $N_2O$?

Eq. 1 is derived from the data in Freing et al. 2009. However, the numbers stated in that paper as the fit to their data are in error, so we here provide the correct numbers as provided by Alina Freing in a pers. comm.. We initially used the numbers exactly as given to us by Alina Freing, but it's true that the number of significant digits is larger than is warranted and we've reduced the significant digits to 7 or 8, so that pN2O is accurate to 2 decimal places.

We added monthly atmospheric measurements at 12 latitudes. Because the observations were not accurate enough prior to 2000 to show a consistent latitudinal gradient and seasonal cycle, the gradient and seasonal cycle were calculated from the data from 2000-2016 and then added to the older global average observations. We added this description to Section 2.6:

"we also ran a series of simulations with the NOAA pN2Oatm observational data that included seasonal and latitudinal variations. Between 2000 and 2014, we used the monthly observations for the 12 available latitudes. Monthly anomalies relative to the global average were calculated at each available latitude from the 2000-2016 observations. These were added to Eq. 1 from 1965 and 1976, and to the global average observations between 1977 and 1999. In the model simulation, the data were linearly interpolated between the 12 latitudes and monthly observations."

And we added to Section 3.2:

"When we used observed atmospheric pN2O that varied with latitude and month (see Section 2.2) the result was essentially the same, with an $N_2O$ flux of 2.4 ± 0.3 Tg N y-1 for the diagnostic sub-model and 2.6 ± 0.3 Tg N y-1 for the prognostic sub-model (data not shown)."

Although not included in the manuscript, we here include a modified Fig. 11 (was Fig. 10 in the submitted manuscript), with the additional simulations using the NOAA pN2Oatm observations shown as crosses (at two low O2 production rates for the diagnostic model and

at the optimum net low O2 production rate for the prognostic model), which shows that when we used the observed atmospheric pN2O the results were essentially the same:

[Figure]

See also the reply to the comment by reviewer 2 on Line 219-220.

I think there might be a mistake in equation 2. Otherwise, I do not see how a value of 2 could mean that the model deviates form the observations by a factor of 2 in either direction. 10^(10log2) = 1024 and it is nothing close to a value of 2. Please explain.

Perhaps, the standard mathematical notation (summation and the number of observations n rather than "average") would be more appropriate here.

The 10 before log indicated that it's the 10-base logarithm. This has been corrected to $\log_{10}$, $\log_{10}(2)=0.31$ and $10^{0.31}=2$. We've converted the manuscript to Latex, which allows a subscript inside a superscript, which makes this distinction more clear. We've changed the formula to $\Sigma\ldots/n$.

It would be useful if the N2O flux calculation in section 2.7 is explained in a little more detail, rather than stating that it "is calculated with the piston velocity from Sweeney et al. (2007)." I am not familiar with this calculation and would love more explanations on how the ocean N2O flux is estimated but the Sweeney et al. (2007) is not listed in the references either.

We've added the equation for the N2O flux calculation, including the piston velocity and the reference to Sweeney:
"N2O flux (=air-sea gas exchange) is calculated as:
    N2O flux = (pN2Oatm*K0*(1-p_watervapor)-
pN2O)*piston_velocity*{660/Schmidt_number_N2O}$^{0.5}$*(1-ice_cover)

, in which K0 is the solubility {WeissPrice80}, p_watervapor is the water vapor pressure {Sarmiento92}, piston velocity = 0.27*(wind speed)$^{2}$ {Sweeney07}, which is optimised for use with the NCEP reanalysis data used here, the Schmidt number for N2O was taken from {Wanninkhof92}, and the ice cover is calculated by the sea ice model LIM2."

I am not sure how equation 3 is used to determine the global air-sea flux of N2O that best fits the ΔpN2O data, if RSS/RSSmin just depends on the number of observations. I do not understand how different model simulations would have different values of RSS/RSSmin if the number of observations is the same.

It is not n that varies but rather RSS varies as the results of different model simulations are compared to the same observations. We added information about regridding and the calculation procedure in Section 2.3, Eq. 4 in Section 2.8, and added clarifications to the legend of Fig. 9:

"The 1σ confidence interval, where RSS equals the value calculated from Eq. 3, is indicated by the horizontal lines. A) diagnostic submodel, each point represents a simulation with a different low O2 slope, B) prognostic model, "no c" is with no N2O consumption i.e. net

production = gross production. All other lines have a constant gross production, and net production varies with different N2O consumption rates. Range of parameter values is given in Section 8.7 of the supplementary material." and of Fig. 11: "MSE^0.5 for the two N2O submodels compared to the $\Delta$pN2O database as a function of global N2O flux at different (net) N2O production rates in the low O2 regions. A) diagnostic submodel, the four lines represent the four best low O2 production rates from Fig. 9A, each point represents a simulation, different symbols indicate different low O2 slopes, points with the same symbols have different oxic N2O production slopes. B) prognostic submodel, the four lines represent the optimised net production rates at the four best gross production rates from Fig 9B, points with the same symbols have different N2O slopes for nitrification."

As for equation 4, I think that its application should be described within the methodology section, rather than just mentioning a little in the discussion section.

Since the F-test at large sample size is insensitive to non-normal distributions we have deleted the equation and accompanying text.

Also, how did the authors optimize various model parameters? And is it not a problem that the optimized oxic $\Delta N_2O$/AOU slope of 12.7 $\mu$mol $N_2O$ (mol $O_2$)$_{-1}$ is so different from the global average given earlier in lines 77-78 (81.5 $\pm$ 1.4 nmol/mmol)? What is the value for this parameter in the prognostic model?

The observed slope of 81.5 $\mu$mol $N_2O$ (mol $O_2$)$^{-1}$ in figure 3 is a weighted average of the low $O_2$ slope and the oxic slope. The optimised slopes in the model are 1700 $\mu$mol $N_2O$ (mol $O_2$)$^{-1}$ under low $O_2$ and 12.7 $\mu$mol $N_2O$ (mol $O_2$)$^{-1}$ under oxic conditions. For the weighted average of these model slopes to equal the observed slope of 81.5, the fraction of $N_2O$ that is produced by the low $O_2$ regions would need to be 4.1% (=(81.5-12.7)/(1700-12.7)). This is close to the 6% for the diagnostic model and 4% for the prognostic model that we find. Since this 4.1% is simply a sanity check that the optimised model does a reasonable job of reproducing the data, but is not an independent estimate, we have not added this calculation to the paper.

The slopes for the prognostic model are given relative to the substrate for each pathway ($NH_4$ for nitrification, $NO_3$ for denitrification). To allow for an approximate conversion to $O_2$ specific slopes (i.e. under the simplifying assumption that $NH_4$ and $NO_3$ are consumed at the same place where they are produced), we've added to section 2.5 that: "Phytoplankton (and all other organic matter) have a fixed C:N:O$_2$ ratio of 122:16:-172." From this it can be calculated that the prognostic model oxic slope of 123 $\mu$mol $N_2O$ (mol $NH_4^+$)$^{-1}$ approximately converts to 11.5 $\mu$mol $N_2O$ (mol $O_2$)$^{-1}$. Because reviewer 3 was not entirely clear whether denitrification in the model is actual denitrification using NO3, we did not add this O2 based slope in the manuscript, as it would add to the potential confusion.

Minor comments

1. L. 24 "It also currently" а・ "It is also currently"

Changed.

2. There are several places in the text, where more detailed or clearer explanations would help readers understand the paper better. For example, l. 53-56 is unclear what the sentence means. Do the authors mean that $\Delta N_2O$/AOU slope becomes negative under suboxic conditions and that leads to the ambiguity of how much $N_2O$ is produced in this region? Please clarify.

We've expanded on the ambiguity to clarify potential reasons for it.

3. L. 71 "observationally derived" a・ "observationally-derived"

The Chicago manual of style says not to hyphenate adverbs ending in -ly.

4. L. 75 Since not all readers of this paper are experts in oceanic biogeochemistry, it would be helpful to explain that the f-ratio is the fraction of total primary production by nitrate.

Added.

5. L. 79 What is the "-$O_2$:C ratio"? What is the dash for?

We've added that "(the - sign indicates that O2 is consumed as CO2 is produced)"

6. L. 233 "N cycle based" a・ "N cycle-based"

Changed to N-cycle-based.

7. L. 242-246 "This estimate…" run-on sentence and needs to be rewritten.

This was split into two sentences.

8. L. 263-267 "It should also…" run-on sentence and needs to be rewritten.

This was rewritten and split into 4 sentences.

9. L. 286 "140 pm" a・ "140 ppm"

Changed

10. L. 290-294 "On the one hand…" run-on sentence and needs to be rewritten.

This was split into two sentences.

11. N-cycle data database used in this paper are shown as embargoed in the data source pointed by the authors (https://www.uea.ac.uk/green-ocean/data). Will the data be publicly available?

The data have now been made publicly available.

---

## Author Comment (AC2) · 25 Oct 2017

I strongly support the goal of this paper, to better constrain the oceanic N2O flux using optimization techniques based on a compilation of datasets of N2O and related N cycle variables, combined with process based models. However, the methodology is difficult to follow and it is hard to believe that all 4 data-based approaches converge to basically the same answer and have the same relatively narrow range of uncertainty, which is governed primarily by uncertainty in piston velocity. There is also no overall sense of what sets this paper apart from earlier efforts, since it seems to be based heavily on what is largely the same delta pN2O data set used before. While I support publication in principle, I think there are many details that should be clarified and explored before this paper is ready for publication.

We thank the reviewer for the comments. We have tried to clarify the methodology throughout the manuscript.

Although four databases were used in our paper, two of these databases, the nitrification rate and the NH4 concentration, were used to formulate a model that was as realistic as possible. They were not used to calculate the N2O flux. We were gratified that the two other databases of depth resolved N2O concentration and surface ΔpN2O converged on the same answer of a low contribution of N2O production in the low O2 region. We have rewritten the end of the introduction to clarify how the four databases are used.

While the literature on N2O fluxes is growing, the only earlier estimate of global N2O flux based on ΔpN2O was by Nevison et al. 1995, 22 years ago, before the MEMENTO database was available. It used interpolation rather than a mechanistic model to obtain fluxes where there are no observations. Therefore our analysis goes beyond existing publications and uses a larger and more complete dataset and process modelling.

Specific comments

L24 Typo: It (is) also currently estimated as the dominant contributor

Corrected.

L32 It's worth mentioning (up front) that this wide range is governed in large part by the possibility of very large coastal and estuarine fluxes. Later on line 87 we learn that the dataset resolution used here is 1x1 degree or 1.1 x 2 degree (plankTOM10.2, line 164), i.e., probably not good enough to resolve these coastal areas.

We have commented on the ambiguity about whether emissions from estuaries are included in oceanic emissions or not in the introduction:

"Part of the uncertainty in the oceanic emissions is whether estuaries are included, which could emit as much as 2.3 - 3.6 Tg N y-1 (Bange et al. 1996)."

We have added a calculation of the contribution of coastal seas, the deep offshore and East equatorial Pacific oceans to N2O flux in Section 3.2 and the Discussion (Table 2 and associated text).

We have expanded on the information about coastal seas and estuaries at the end of the discussion:

"The largest coastal seas are resolved in our model although the processes related to specific coastal environments are not, such as the interactions with sediments and with tides. Our results do not include emissions from estuaries."

L43 although there are additional pathways, such as (please give brief list).

We initially wrote it like this because the reference for this statement, Klawonn et al. 2015, measured a large number of N-transformation reactions, so that a list would not be brief. However, to respond to the reviewer's comment that the statement needed more detail, we have replaced it with ", although denitrification may be significant in the anaerobic centres of

large marine snow particles in oxic waters", because that is the most important pathway identified by Klawonn et al.

L72 probably should mention up front that the deltaN2O/AOU data are based on MEMENTO. Otherwise, it's a bit confusing to know the basis of this calculation.

This information was moved from the Fig. 3 legend to the main text: "The globally averaged ΔN2O/AOU ratio was calculated from the MEMENTO database (Bange et al. 2009) as 81.5 ± 1.4 µmol/mol (Fig. 3)."

L76 NPP is estimated at 58 +/- 7 PgC/yr based on what? An ocean model? Satellite data? Even at the lower end of 51 PgC/yr, this is on the high side of satellite-based estimates.

This estimate is based on our previously published work (Buitenhuis et al. 2013a). It uses the same methodology as in the present paper. We have added the following text to clarify the origin of the estimate: "based on $^{14}$C primary production measurements (n=50,050), parameter perturbations of a previous version of the model used here, and Eq. 3". Our estimate is broadly within satellite algorithm estimates, which range from 38-70.7 Pg C/y. We have found that the vertically integrated primary production from our model reproduces the observations better than the best satellite algorithm. It also had the second lowest error (root mean square difference) of vertically integrated primary production relative to observations in the Arctic only out of 21 biogeochemical models (Lee et al. 2016 doi:10.1002/2016JC01193). Depth resolved primary production constrains global NPP even better than that at 58 +- 7 PgC/y., as discussed in Buitenhuis et al. 2013a.

L83, please list relevant references rather than just saying "(see Introduction)".

We have replaced this by a reference to Fig. 4 (was Fig. 11) which includes all the references.

Line 92-93. Since pN2O is close to equilibrium in much of the ocean, it seems important to consider the quality of these pN2O measurements. For example, surface measurements made with underway systems tend to have much higher precision than analyses based on bottle collection. Was the uncertainty comparable across the MEMENTO database and if not, were the differences in data quality accounted for in the subsequent calculations?

Annette Kock (who does the technical support for MEMENTO in the group of H. Bange) informally estimates that more than 95% of the surface pN2O data entries included in MEMENTO have been measured with underway systems. In addition, comparison experiments between underway and discrete measurements show an overall good agreement between both methods (Arévalo-Martínez et al., 2013, doi: 10.5194/os-9-1071-2013). We have added a caveat to the manuscript: "Since there is at present no formal quality control beyond that performed by individual contributors to the MEMENTO database and a check by the database administrators that the values make physical sense (Kock, A., and Bange, H. W.: Counting the ocean's greenhouse gas emissions, Eos, 96, 10-13, 10.1029/2015EO023665, 2015), we have taken the MEMENTO database at face value."

L125 Ocean models often do a poor job of reproducing observed O2. Suntharalingam 2012 used WOA O2 rather than model O2 for that reason. Presumably, the sensitivity to light, temperature and O2 described here is based on values from plankTOM10.2 (if not, please clarify). How well does plankTOM10.2 reproduce O2 relative to observations? (Note: I saw later that my question was addressed in the Results on lines 184-188. That material belongs up front in the methods description.

We have added this information to section 2

"As will be described more fully in Section 3.1, we used observed O2 concentrations in the simulations (Bianchi et al. 2012) rather than interactively modelled O2, to minimise the impact of model biases in simulated O2 fields (Suntharalingam et al. 2012)."

L132 paragraph starting here. This paragraph could be written more clearly, especially the sentence spanning L137-138. What is a variable N quota? Is the model using Michaelis Menten kinetics?

The paragraph was reworded, and references were added that provide further documentation of the model formulation and the contrast between a quota model and a Michaelis Menten kinetics model:

"The model uses a fixed C:N:O2 ratio for all organic matter of 122:16:-172, and Michaelis-Menten kinetics for growth rate based on inorganic N uptake by phytoplankton (Buitenhuis et al. 2013a, supplementary material Eq. 8, 9). We therefore need a K1/2 for growth rather than for uptake to be consistent with the fixed C:N ratio (Morel 1987)."

On line 144, a low cost function of 3.3 is better than the cost functions of >4 described for the previous model, correct? Yet, the sentence beginning on L142 with "However" suggests a large uncertainty.

This is indeed confusing (yes, 3.3 is better than >4), and it was rewritten to give a more equal balance between the small scale unexplained variability and the large scale pattern that is well reproduced by the model:

"Due to the highly dynamic nature of NH4+ turnover, the ability of the model to reproduce the observed NH4+ concentrations at the same times and places was by no means perfect, but the large scale pattern of surface NH4+ concentration shows an increase with latitude, consistent with the observations (Fig. 6), which translates into a cost function of 3.0."

L155-156 What is meant by "The slopes of the three processes" ?

We have changed this from slopes to ratios, and added an explanation at the end of the second paragraph of the introduction:

"Throughout the manuscript we will refer to N2O stoichiometries relative to O2, NH4 and NO3 as ratios, because they have been optimised against global databases of concentration measurements, rather than from microbiological yields. Using the latter would be more mechanistically satisfying, but the relevant yields are at present insufficiently constrained by observations."

Section 2.2-2.8. General question. Do the 4 databases described in section 2.2 correspond to the specific sections 2.4-2.7? If so, where does section 2.8 fit in? Is Equation 3 an alternative cost function to Equation 2 described in Section 2.3?' The apparent switch from Equation 2 to Equation 3 as the optimization technique is confusing and unclear.

The 4 databases correspond to sections 2.4, 2.5, 3.1 and 3.2. It was split in that way because the nitrification and phytoplankton NH4 use are necessary model developments before we can implement the prognostic model, but they are not part of the main result of the paper, which is an estimate the ocean-atmosphere N2O flux and its confidence interval. The switch between Eq. 2 and 3 is split in the same way. We've been using Eq. 2 in multiple previous publications because it gives equal weight to biases in small and large numerical values, and is therefore is therefore more appropriate for optimising a global model that spans a range of conditions. We have added this to Section 2.8:

"In previous versions of the PlankTOM model (Buitenhuis et al. 2006, Buitenhuis et al. 2010, Buitenhuis et al. 2013a) we have used Eq. 2 to evaluate the model because it minimises relative error, which we have found to be more appropriate when the observations span several orders of magnitude. Unfortunately, statistical confidence intervals have only been defined for $\chi^2$-statistics such as Eq. 3 and 4, which minimise absolute error, so that we end up with two cost functions (Eq. 2, 3), depending on the application."

Line 190 and Figure 6. The model substantially underestimates N2O in the most important hotspots of production. Doesn't this mean it will tend to underestimate global N2O production?

It seems like this concern is dismissed somewhat casually with handwaving arguments, e.g., the text starting on line 289.

We have given considerable attention to the underestimation of N2O at depth in the low O2 regions, and discuss it from different angles in paragraphs 3, 6 and 7 of the discussion.

We have added Table 2 and a paragraph in Section 3.2 (second paragraph, starting "High N2O fluxes"), that analyses the contribution of N2O hotspots to global N2O flux.

We have rewritten paragraph 3 (starting with 'This lack of knowledge") of the discussion to more explicitly present the balance of evidence whether or not the underestimate of N2O concentrations at 500-1000m depth (Fig. 7) influences N2O flux at the surface. On balance, including the new Table 2, our analysis still suggests a small global significance of the hotspots. This conclusion was also reached by  Freing et al. (2012), and we added a reference to this: "Freing et al. (2012) also estimated a small fraction of 7% of the global total contributed by denitrification/low O2 N2O production."

Line 201-202 Please clarify how these results link together. Line 201 says that both hypoxic production AND CONSUMPTION were optimized. The subsequent results mention GROSS production of 0.33 TgN/yr, then optimized N2O production of 0.12 or 0.16 TgN/yr. Are the latter results net production? Can we infer that about 0.17- 0.21TgN/yr is consumed in suboxic zones?

Yes the results are for net production and we have added the word net to clarify this. Yes, the optimised consumption is 0.21 Tg N/y. We have also added clarifications throughout the manuscript on which processes occur in suboxic, hypoxic and oxic waters.

Line 202-203 Total production of 0.12-0.33 TgN y-1 in low O2 is only about 10% on average of total production. This is much lower than the 33% suggested by Suntharalingam et al. 2012. Does this mean that the authors are concluding that the OMZs are only responsible for a small fraction of global N2O production? Please expand on this point and call it out more explicitly in the Discussion.

We are not confident that the lower attribution to denitrification produced by our current model version is better than published by Suntharalingam et al 2012. This is detailed in the Discussion, to which we've added references:

"Both the diagnostic and the prognostic models assign a small percentage of the total N2O production to the denitrification pathway, 6 and 4% respectively. However, because of the large bias between the observed and modeled N2O concentration depth profiles (Fig. 6) these may be underestimates (Suntharalingam et al. 2012, Arevalo-Martinez et al. 2015)."

We have revised the text to discuss the possible implications of this shortcoming, which we argue do not significantly affect our results for the total global N2O flux. See our reply to the comment above on Line 190 and Figure 6, that outlines changes we made to Section 3.2 and paragraph 3 of the Discussion to more clearly present the balance of evidence.

Line 204 perhaps add a clause clarifying that the .0017 molN2O/molO2 slope is about 20 times the mean deltaN2O/AOU ratio of 8.15e-5 from line 82.

The .0017 slope is the gross production slope in the low O2 regions. The 8.15e-5 slope is the net slope averaged over the whole ocean. It is therefore to be expected that the former is larger than the latter. See changes made in response to the next two comments.

Line 205 production for the prognostic model is given in units of umol N2O (mol NO3)-1. Does this represent NO3- consumed by denitrification, or produced by nitrification? Can you provide an estimate of how this relates to the previous units of mol N2O/mol O2?

We have added clarifications on which slopes apply to denitrification and to suboxic, hypoxic and oxic waters. Since NO3- consumption and O2 consumption are spatially separated, stating a

N2O/O2 slope would be confusing, but we have added the model N:O2 ratio in Section 2.5, so that the magnitudes can be placed in context.

Line 216 Please use consistent N2O (mol O2)-1 slope units. Here the units are umol/mol. On line 78 they were nmol/mol. On line 204, they were mol/mol.

All slopes have been converted to a denominator in mol, and a numerator in µmol, or in mmol if it was >1000 µmol.

Line 217 How does this nitrification slope in units of umolN2O/molNH4+ relate to the

"N2O production slope" on line 206 in units of mol N2O/mol NO3-?

We have added clarifications on which slopes apply to denitrification and to suboxic, hypoxic and oxic waters.

Line 219-220 Are all measurements really of deltapN2O, or are most of pN2O in the surface ocean? In the latter case, what is the uncertainty in pN2Oatm, e.g., from Eq 1?

We have added in Section 2.2 that:

"The average absolute difference relative to the global average pN2Oatm data from the NOAA/ESRL Global Monitoring Division (ftp://ftp.cmdl.noaa.gov/hats/n2o/combined/HATS_global_N2O.txt) is 0.5 ppb between 1977 and 2014 and 0.3 ppb between 2000 and 2014." See also the reply to the comment by reviewer 1 on equation 1, in response to which we ran the optimisation using the NOAA/ESRL data, and got essentially the same results.

Line 220 On what basis was this 1978 estimate made? Is there updated information that could be used?

It is cited in Cohen and Gordon 1978 as a personal communication from Weiss, who calculated the solubility we used based on published data. No further details are given, but because it's an order of magnitude less than the uncertainty in the piston velocity, this doesn't strike us as a problem. We are not aware of more recent measurements. The Sarmiento and Gruber textbook (2006, Ocean biogeochemical dynamics, ISBN: 9781400849079) also gives the solubility we used as the most up to date one.

Line 233 Typo or confusing sentence.

Based on the comments of reviewer 1, we hyphenated "N-cycle-based".

Line 248-250. It would be good to provide references to support these claims.

We have added references to the C-cycle data, and refer to the figures with observational data for the N-cycle data.

Line 282 – paragraph starting here. This exercise, combined with large data gaps in Figure 9a, including in both the ETSP and ETNP, suggest to me that the authors are overstating the degree of certainty in their confidence interval. There are large areas of the ocean with no data, including in the most important hotspots for N2O production.

The observations are in fact fairly evenly spread (Table 2). There are observations in the ETSP and ETNP. The observations include upwelling regions. The analysis we present of a hypothetical undersampling of high values suggests that constraining the piston velocity would narrow the confidence interval more than making more pN2O measurements. As mentioned in response to the reviewer's comment on L32, we have added an analysis of N2O flux in coastal seas, the deep offshore and East equatorial Pacific oceans to quantify the contribution of these hotspots.

Line 303-304. This is the first mention of the fact that the model produces low fluxes from the Southern Ocean. Can you cite the relevant figure here and call attention to this point earlier in

the Results section? (Figure 9b,c,d all seem to indicate a substantial flux from the Southern Ocean.)

It is the atmospheric inversions that produce low fluxes in the Southern Ocean, rather than the process models presented here. We have rewritten these sentences to make this clearer:

"However, N2O emissions from inversions in the Southern Ocean are lower than the priors (Hirsch06, Huang08, Thompson14, Saikawa14). These low Southern Ocean emissions (0.02 – 0.72 Tg N y-1) are consistent with our results (0.68 – 0.79 Tg N y-1). South of 30S, 88% of the Earth surface is ocean, resulting in a clearer attribution in the inversions of the atmospheric N2O anomalies to ocean fluxes. We suggest that the higher emissions estimates from inversions could be due to a combination of overestimated priors of ocean fluxes in combination with insufficient observational constraints at latitudes North of 30S to allow correct partitioning between land and ocean fluxes."

Line 308-310. The neglect of estuaries is indeed a key uncertainty, which needs to be mentioned much earlier, i.e., in the Introduction. It is also debatable whether coastal areas are adequately represented in the models presented here, which 2x1 or 1x1 resolution.

See reply to question on L32.

L487 Figure 2. This figure suggests very high f-ratios, e.g., of 0.8-0.9 in the northern subtropical gyres, that are a little hard to believe. The global mean looks to be on the order of 0.4! Are these generally accepted values or are they biased by measurements in highly productive coastal waters?

This one point of 0.8 is the average of 2 measurements (0.8 ± 0.1) made at 14N21W and 21N21W, about 4° West of Africa (Varela et al. 2005). In the Discussion we point out (first paragraph) that the f-ratio (global mean = 0.29 ± 0.18) is the largest contributor to the uncertainty in N2O production we estimate from the N-cycle-budget. We also explicitly state that further (synthesis of) observations would help constrain this uncertainty. We think that trying to ensure a representative mean by weighting some values more (e.g. open ocean measurements) would be too subjective / sensitive to the exact methodology used.

L505, "Model results are for the same months and longitudes as the observations." What about latitudes?

We have added the clarification to the panel D legend:

"Latitude y-axis to the left of panel A."

L527 This plot is dominated by the error bars and somewhat obscures the focus on the mean value, which arguably is the more important quantity. The current study makes a much more detailed effort to quantify uncertainty than most of the previous studies (some of which make no effort at all). Could a separate panel with a narrower Y-axis range be plotted to better compare the mean value of the fluxes? And can you please provide some discussion of the main factors contributing to the differences in mean value?

Error bars are important when comparing different estimates. We have decreased the aspect ratio of the figure to 1, so that differences along the y-axis become easier to read.

We have added a discussion of the two main factors contributing to the different N2O flux in Nevison95:

"Because of differences in methodology it is not possible to provide reasons for why our estimate is lower than the more recent estimates. We can, however, compare our estimate to that of (Nevison95), because it is also based on a database of ΔpN2O. Compared to their high end estimate using the piston velocity of Wanninkhof of 5.2 ± 3.6 Tg N y-1, our estimate is lower because we use the more recent 13% lower estimate of piston velocity of (Sweeney et al. 2007), and because our ΔpN2O of 7.6 ± 18.1 ppb is 25 - 28% lower compared to 10.55 natm in Nevison

(1995) (the range is calculated based on the water vapor correction for conversion between ppb and natm, which increases from 0.6 - 4.1% at temperatures from 0 - 30 °C, which brings the values slightly closer together)"

---

## Author Comment (AC3) · 25 Oct 2017

Thanks for this contribution to global marine N2O modeling. May I ask some questions regarding the model formulation and applied parameter sampling:

Line 148: Is there nitrification at 1 umolO2/l?

Almost none in our analysis. Nitrification declines in the same way with O2 as remineralisation switches from O2 to NO3. We added: "Since nitrification consumes O2, in the model it decreases as remineralisation switches from O2 to NO3 (supplementary material Eq. 70, 61, 67)."

Line 156:  How is N2O consumption modeled?  As a first order consumption term as applied in other studies?  How large is gross consumption?  What O2 threshold do you use to separate nitrification, production from denitrification and consumption from denitrification?  How large are aerobic and anaerobic remineralization fluxes in the model?

We added to Section 2.6:

"The ratios of the three processes are globally invariant (supplementary material Eq. 70, 61, 63, 71). The functional form of the O2 dependence of N2O consumption (suppl. Eq. 71) was the same as that of denitrification (suppl. Eq. 67), and with an O2 response function that is 1.5 µmol $L^{-1}$ lower than that of denitrification, which is similar to that used by Babbin et al. (2015). We independently optimised the ratios of N2O production and consumption from denitrification (Section 3.1), which controls the net N2O production as a function of O2 concentration. There is not enough information at present to optimise the O2 concentration parameters of denitrification and N2O consumption as well."

We have added the full set of equations for nitrous oxide to the model description as supplementary material, with references to the relevant equations throughout the Materials and Methods section. Optimised gross consumption is 0.21 Tg N/y, see the answer to reviewer 2 question on Line 201-202. The O2 thresholds for N2O production (34 µmol O2 L-1) and consumption (28 µmol O2 L-1) are stated in Section 3.1, the O2 threshold for nitrification is 0 µmol O2 L-1 (see previous comment). Primary production is 64.5 Pg C/y, of which 99.5% is remineralised aerobically (using 7540 Tmol O2/y) and 0.008% is remineralised by denitrification (using 0.485 Tmol NO3/y). The rest is partitioned between removal of nutrients at the sediment to compensate for nutrients added by rivers (da Cunha et al. 2007, GBC) and changes in the inventory of total organic matter.

Line 166: Are modeled N2O concentrations not drifting substantially after such a spin

The length of the spin-up is a trade-off between keeping the runs short enough that the nutrient distributions are close to observed ones, so that the model behaves realistically when it is formulated and parameterised using physiological and ecological observations (Buitenhuis et al. 2006 Global Biogeochemial Cycles, 2010 Global Biogeochemial Cycles, 2013a, Le Quere et al. 2016), and long enough that the N2O concentration and ΔpN2O distributions can be used to optimised N2O process rates. Our optimized model is by definition as close as possible to observations, even if the deep ocean is not fully at equilibrium. With this method we were able to conduct a large number of model experiments, a sub-ensemble of which are presented here.  We note in Section 3.1 that N2O production below 1600 m, where there is an increase in concentration, is only 5% of the total production. Given the slow ventilation of the deep sea, this accumulation will have a negligible effect on the optimised flux, and keeping the simulation short actually helps with this.  We have added an analysis of the effect of variability to Section 3.1 and corresponding text as follows: "The results were the same in both diagnostic and prognostic submodels for the 2000-2004 and 2005-2009 averages, showing that the model was sufficiently spun up."

Line 199:  How many parameter perturbation simulations did you run?   Which sampling technique is applied to vary parameters?  Over which range are parameters varied?  What does the legend in Fig 8/10 stand for?  Could you illustrate the sampled slopes and resulting optimal slope?  Are fluxes tied stoichiometrically to remineralization fluxes?   Why is N2O consumption slope given as N2O/NO3-?  Does this make sense stoichiometrically?

We did a large number of parameter perturbations, 6 of which were used to constrain the low O2 N2O production in the diagnostic model (Fig. 9A), 23 to constrain the low O2 N2O production in the prognostic model (Fig. 9B), 27 to constrain total N2O flux in the diagnostic model (Fig. 11A) and 26 to constrain total N2O flux in the prognostic model (Fig. 11B). Parameters were varied until they constrained the optimised rates and their confidence intervals. We've added to the Fig. 9 legend that "each point represents a simulation with a different low O2 slope". We clarified the text following Eq. 2 : "To calculate the cost function (and also to calculate RSS in Eq. 3), the model was regridded to the same grid as the observations, and residuals were calculated at months and places where there are observations." We added to the Fig. 9 legend: "Range of parameter values is given in the supplementary material Section 8.7" Based on the question of reviewer 3 about Eq. 3 we have clarified the legend of Fig. 9 (was Fig. 8). Section 8.7 of the supplementary material also gives the optimal slopes, which are also given in Sections 3.1 and 3.2. N2O consumption occurs because N2O is consumed during denitrification. This was added in Section 3.1. NO3 is the substrate of denitrification, and N2O is an intermediary, so it does make stoichiometric sense that N2O can act as an alternate substrate for denitrification.

Figure 6: Many global N2O modeling studies present N2O versus O2 scatter plots for evaluation. What does this relationship look like in the model? The N2O flux estimate of 2.4+/-0.8 Tg N yr-1 is much lower than what was reported in Suntharalingam et al. 2000/2012, on which the model builds ('4.6 Tg N yr-1 (comprised of 3.0 Tg N yr-1 from the 'nitrification' pathway, and 1.6 Tg N yr1 from the low-oxygen pathway', Suntharalingam et al.  2012).  How come?  Does your prior include these previous fluxes?  Your N2O production at low O2 is now ~10 times smaller compared to this previous model.

We present the modelled N2O and observed N2O next to each other in Fig. 7. Since this shows N2O as a function of depth for different basins, this has a higher information content than a scatter plot.

x-axis O2(µmol L-1), y-axis N2O(nmol L-1), black observations, green diagnostic model, red prognostic model.

The observational O2 are used in the model, so this plot does not add any information relative to Fig. 7, and we have not included it in the manuscript.

We have added a discussion of the two main factors contributing to the different N2O flux in Nevison95:

"Because of differences in methodology it is not possible to provide reasons for why our estimate is lower than the more recent estimates. We can, however, compare our estimate to that of (Nevison95), because it is also based on a database of ΔpN2O. Compared to their high end estimate using the piston velocity of Wanninkhof of 5.2 ± 3.6 Tg N y-1, our estimate is lower because we use the more recent 13% lower estimate of piston velocity of (Sweeney07), and because our ΔpN2O of 7.6 ± 18.1 ppb is 25 - 28% lower compared to 10.55 natm in Nevison (1995) (the range is calculate based on the water vapor correction for conversion between ppb and natm, which increases from 0.6 - 4.1% at temperatures from 0 - 30 °C, which brings the values slightly closer together)"

This is the only estimate where the methodologies are comparable enough (based on an observational database of ΔpN2O, and using a piston velocity that is a function of the square of the windspeed) that we can isolate quantitative reasons for the differences in the estimates.

This is not an inversion, so there's no prior.

---

## Author Comment (AC4) · 25 Oct 2017

The manuscript by Buitenhuis and Coauthors describes the results of simulations with an ocean biogeochemical model that includes different parameterizations of N2O production, constrained with available N2O observations. The main finding is a net N2O outgassing to the atmosphere of 2.4 ± 0.8 TgN/year, which is substantially lower than many of the estimates previously reported, and also less uncertain. A very small proportion of the N2O production comes from denitrification-associated pathways in suboxic waters. The estimate also appears robust to the choice of the parameterization of N2O production.

This is a short and potentially useful paper, although not particularly original. But I think that, if better supported, the results will push other scientists to reconsider estimates of N2O emissions from the ocean (as well as from other sources) in light of the low values suggested. That said, I also think the paper is poorly written, in particular when it comes to the description of the methods employed - for example the model equations, the rational and choices behind the parameterizations, the steps behind the optimization. Furthermore, I have some additional concerns about the results that prevent me from fully supporting publication of the manuscript as is.

We thank the reviewer for the comments. We have tried to clarify the methodology throughout the manuscript.

Specific comments:

- The model formulation is quite hard to follow, partly because equations are not show. This makes it difficult at times to judge the validity of the model's assumption. I strongly encourage the Authors to show all the pertinent equations, either in the main text, or in an appendix.

We have added the full set of equations for nitrous oxide to the model description as supplementary material. This has all the equations, parameter values, and how it is set up. We have also made multiple clarifications in the text following the reviewer's comments. Please see point-by-point responses below (and in response to the other reviewers).

- The choice of some of the model equations and parameterization is unclear and could be better justified. The Authors could do a much better job explaining why certain functional forms have been utilized, and what consequences these choices may have, if any. For example, looking at Table 1, line 3 lists an equation that uses the logarithm of O2. Is there any reason for this form? The logarithm will expand the range at very low O2 concentration - do we trust O2 measurements there? Further, this form breaks down at O2=0; does this ever happen in the model/observations, and is there any limit applied to prevent it? Finally, all of these equation should represent a process ultimately limited by O2. Is there any limitation as O2 goes to zero?

On the specific justification for the use of logarithm of O2, the choice is due to a better fit to the data. We now explained this in the text:

"A logarithmic function fit the data better than linear, exponential, or power functions."

We have also added an explanation why the N2O consumption equation and parameters were used:

"The functional form of the O2 dependence of N2O consumption (suppl. Eq. 71) was the same as that of denitrification (suppl. Eq. 67), and with an O2 response function that is 1.5 µmol L$^{-1}$ lower than that of denitrification, which is similar to that used by Babbin et al. (2015). We independently optimised the ratios of N2O production and consumption from denitrification (Section 3.1), which controls the net N2O production as a function of O2 concentration. There is not enough information at present to optimise the O2 concentration parameters of denitrification and N2O consumption as well."

The choice of model equations for the preferential algal uptake are justified in Vallina and Le Quéré (2008).

The lowest O2 concentration in the Bianchi et al. 2012 database (after regridding onto the model grid) is 1.15 μmol O2/L. The lowest concentration for which there is a yield measurement is 5.4 μmol O2/L. It is therefore true that the logarithm extrapolates the N2O/AOU ratio from 232 μmol/mol at 5.4 μmol O2 to 251 μmol/mol at 1.15 μmol O2. Given the variability in the measurements this is an insignificant extrapolation beyond the range of the measurements. Also, nitrification rate decreases with O2, so that the N2O production rate is low.

- I found the distinction between the prognostic and diagnostic model (for N2O) somewhat confusing.  In both models N2O is carried as a prognostic tracer - except in the first model it does not depend on other N-cycle tracers, and is not consumed, but only produced and passively advected until it outgassed from the surface. What makes one model diagnostic and another prognostic?

In this manuscript we use the distinction between diagnostic and prognostic model to mean that the former is based on statistical relationships with observations, while the later is based on process understanding and representation. The N2O field from models are indeed transported in the same way. We clarified this in Section 2.6 as follows:

"N2O production is implemented as two distinct submodels. The diagnostic submodel is based on statistical relationships of DeltaN2O/AOU ratios taken from observations and has previously been published {Suntharalingam00,Suntharalingam12}.",

"The prognostic submodel presented here is based on process understanding and explicitly represents the primary N2O formation and consumption pathways associated with the marine nitrogen cycle (Fig. 1)."

and "The N2O concentrations from both the diagnostic and the prognostic model are transported in the same way by physical transport and the formulation of their gas exchange is also identical."

- Regarding the prognostic model - the Authors say that it explicitly represent the redox transformations that lead to the conversion of NH4+ to eventually N2O (actually only a subset, as for example, the model does not include NO2-), but the model seems to still parameterize them heavily. For example the current understanding is that N2O is an obligate intermediary during heterotrophic denitrification, so that one should expect a gross N2O production comparable to the denitrification rate (i.e. ~70 Tg N/year)

However, the Authors indicate a suboxic gross production of only 0.33 Tg N year – a very small rate in comparison.  This may be explained by the use of "slopes" in the prognostic model that relates N2O to other tracers (more on these slopes in the next comment). This implicitly assumes a tight coupling between production and consumption at suboxic levels, with only a fraction of the N cycled by denitrification escaping to the water column.  It may be fine as a parameterization - especially since it is one that is optimized against observations.  However it may be problematic if the model is to be used under varying circulation or climate - the coupling between production and consumption may vary, and given the large gross N2O fluxes this may impact net production and accumulation of N2O.

The model only represents the process of denitrification, it does not represent a state variable for denitrifiers (first sentence of Section 2.4). Therefore, reactions that happen intracellularly in denitrifiers are not represented either, and gross production from denitrification represents N2O production that is exuded/leaks into the surrounding seawater and stays there long enough to leave a measured increase in N2O concentration. The small net production rate is a result of the optimisation against observations, there is no implicit assumption built into the model that production and consumption should be tightly coupled. We don't present climate change simulations here, so we cannot comment on whether using the present model for climate change projections would be more or less problematic than using any of the other available models.

- It is unclear what "slopes" are used in the prognostic model. Are these slopes actual yields (e.g. N2O production per NH4+ oxidized), relationships with O2 consumed, or just empirical relationships based on data syntheses? And what is then the slope of the third step of N2O cycling (consumption of N2O by denitrification)? Is it a relationship with O2, with NH4+ or with NO3-? (and specifically, is there explicit denitrification in the model, so that one could relate N2O production to NO3- deficit?).

We have changed slopes to ratios, and explained our reasoning in using the word ratios rather than yields at the end of the second paragraph of the introduction:
"Throughout the manuscript we will refer to N2O stoichiometries relative to O2, NH4 and NO3 as ratios, because they have been optimised against global databases of concentration measurements, rather than from microbiological yields. Using the latter would be more mechanistically satisfying, but the relevant yields are at present insufficiently constrained by observations."

Yes, denitrification is explicitly represented, as stated in Section 2.4. N2O consumption is therefore proportional to NO3- consumption, as stated in Section 3.1. Yes, it would be possible to calculate a NO3- deficit, such as N*. We judged this to be outside the scope of the paper, because denitrification can be accompanied by both N2O production and consumption, so model validation of denitrification rate against observations of N* would not help constrain the N2O budget. We added supplementary material to this paper which contains a detailed description of the biogeochemistry model equations (taken from Le Quere et al. 2016, and now updated with a description of both N2O submodels (section 6.5 and 6.6).

- The lack of spinup in the model is worrisome: the model was apparently initialized in 1965 and run for 49 years through 2014. This is a short running time, and it completely misses a spinup phase. It may very well be the case that the N2O inventory of the ocean over the last 5 years is still adjusting from the initial condition, in a way that could bias the outgassing estimates. For example, there seems to be a substantial accumulation of N2O in the deep ocean - if this is still ongoing after 49 years, then the outgassing estimated by the Authors could be a lower estimate. A comparison between the total net production in the interior and the outgassing could give a sense of any disequilibrium. Note that a similar modeling study by Martinez Rey et al., 2013, BGS (incidentally finding about 4Tg/year emissions) suggested a 150-year spinup was not enough to eliminate drifts in N2O and other biogeochemical variables. Any drift should be discussed in the paper, and the consequences assessed.

We have added an analysis of the optimised N2O flux for the 2000-2004 and 2005-2009 periods, which show the same result. We note in Section 3.1 that N2O production below 1600 m, where there is an increase in concentration, is only 5% of the total production. Given the slow ventilation of the deep sea, this accumulation will have a negligible effect on the optimised flux, and keeping the simulation short actually helps with this. The frequency distribution of ΔpN2O in the submodels closely matches that in the observations (Fig. 12), which supports the conclusion from the small error attributed to the model-observation ΔpN2O mismatch, that the model does not have a major bias.

Martinez-Rey et al. do climate change simulations, and spin up the model so that they don't have to include control simulations and present the climate impacts relative to the control. Our study is different, where we initialise from the available observations and optimize model parameters using the available observations to derive the present day oceanic N2O flux. See also our reply to the comment of Gianna Battaglia on Line 166.

- I found the description of the optimization steps very unclear. It took me a while to figure out what steps the Author follow and how the model is actually compared to the data, and I'm still not sure about them. Now my understanding is that a first optimization is carried out for the NH4-cycle using nitrification rates and NH4+ concentrations; then a second optimization is performed with interior N2O data to determine parameters for low-O2 pathways (but does this apply to both the prognostic

and diagnostic model?); and finally a third optimization (presumably with some parameters fixed by the previous steps?) using surface Delta-pN2O data for the global source terms, used to determine the final air-sea fluxes. That's my understanding but I am still not sure I got it right, and some aspects remain puzzling. I think this could be much better explained from the start, for example by a method section outlining the optimization strategy in more detail.

We do carry out 3 optimisations, but we split the presentation into two parts, one where we develop the model so that we can implement the prognostic model, and the other where we use the model to optimise the model to the two N2O datasets. We do not include NH4 concentration database in the optimisation because the high turnover rates and the many processes that are involved would make this a process that would require a whole paper by itself, which is outside the scope of the present paper. Fortunately, the many processes turn out to be reasonably well constrained by observations we present in this and previous papers (Buitenhuis et al. 2006, 2010), so that we judge the resulting NH4 concentration distribution to be fit for the purpose of optimising the N2O cycle which we undertake here. We have more explicitly described the progressive steps of how we use each observational dataset at the end of the introduction, from model development of the N-cycle in Section 2 to identifying N2O rates that best fit the observations in Section 3. See also the reply to the first comment of reviewer 2. We have clarified the legends of Fig. 9 and 11, see reply to reviewer 1 comment on Eq. 3. See also the reply to reviewer 2 on Section 2.2-2.8.

- Related to the previous comment, the equations for the optimization are absolutely opaque and unclear. They need to be substantially clarified: ideally anyone should be able to apply them after reading the paper, which is not the case. For example equation (1) is not very specific: instead of "average", "model", "observations" the actual mathematical form could be given - this would also help knowing how the average was done, wether the in situ or gridded data were used, how the model was sampled etc.

We have changed the mathematical form of Eq. 3 and 6 to replace average by the sum divided by the number of observations. We added Eq. 6 to give the actual mathematical form of the model and observation data used. We have added that the model was converted to the same grid as the observations, and sampled where there are observations in Section 2.3. See clarification added in response to Gianna Battaglia's comment on Line 199.

Similarly I am completely at loss with section 2.8, and I could not trace back the steps applied by the Authors based on this description alone. How is RSS/RRS_min (equation 3) used, how does it relate to the quantities shown in Fig. 8 and 10, and why does it only contain the number of observations but no information on the actual variables?

We explained how Eq. 5 (was Eq. 3) relates to Fig. 9 and 11 (was Fig. 8 and 10) in the legends of these figures: "$MSE_{min}$ was obtained as the minimum of a second order polynomial fit (black lines). The $1\sigma$ confidence interval, where MSE equals the value calculated from Eq. 5, is indicated by the horizontal lines." We have added Eq. 6 to show how the actual variables (observations and model results) are included in the calculation of MSE (=RSS/n).

What does the "phi" term (equation 4) represent, and how is it actually used?

Because the paper we discussed only tested sample sizes that were more than 2 orders of magnitude smaller than our database, we decided to delete this equation and text.

- Regarding the final estimate of N2O air-sea flux, I think it could be couched much better into the context of previous estimates (also, a table would help), and what could be behind the potential discrepancies in light of the substantially lower revision. This could be especially interesting given that many modeling studies use a similar approach. The Authors also present an "observational" estimate of N2O production whose central value (4.6 TgN/year) is quite different than the final model estimate - this discrepancy could be added to the discussion. I am not particularly surprised by the lack of sensitivity of N2O production to the choice of diagnostic and prognostic models, since

both are optimized versus observations. Surface pN2O should be a quite powerful constraint to outgassing fluxes.  However, one may still expect different sensitivities to interannual variability and climate change, so this is not a strong argument in favor of not resolving complex pathways that characterize the low-O2 N2O cycle.

We present the context of previous estimates in Fig. 4. We have added a discussion of the discrepancy with the Nevison et al. 1995 estimate using Wanninkhof piston velocity in the 4[th] paragraph of the discussion (paragraph starting "Despite these shortcomings"). Because the methods of other previous estimates are different, we can't give specific reasons why our results are different from the other estimates. The observational estimate in Section 2.1 is similar to (NOT quite different from) the combined model-observation estimate: the confidence interval of that estimate completely overlaps our better constrained estimate in Section 3.2. We have added this to the discussion:

"This estimate of global marine N2O production derived from analyzing the N cycle is statistically indistinguishable from the N2O flux derived from DeltapN2O observations, but has a much larger error."

We note that our estimate of the optimised N2O flux is sensitive to the observational dataset used, but not to the details of the model. Since our model parameters are optimised using a database spanning multiple years, and not on a year to year basis, we note that this model specification is more suited to estimating long-term or climatological fluxes, and not interannual variability.

- The model is biased in its representation of export and remineralization, as well as N2O distribution.  The discussion of the effect of these biases (e.g.  lines 289-300) is not especially thorough - so the conclusion, in particular regarding the narrower range of the new estimate, is not very convincing.  Furthermore, there are hidden resolution biases.  For example, the model can not resolve low-O2 coastal upwelling regions, which have been shown to be powerful conduits to N2O outgassing (e.g.  Arevalo Martinez et al., 2015, Nature Geosciences). The abstract/conclusions could be more cautious with respect to the real uncertainties.

We have rewritten the discussion of the bias due to the too deep remineralisation, to more explicitly present the balance of evidence whether or not the underestimate of N2O concentrations at 500-1000m depth (Fig. 7) influences N2O flux at the surface:

"it should also be noted, first, that the optimization using surface ΔpN2O agrees with the optimization using N2O concentration that the contribution of the low O2 N2O production needs to be low (Fig. 11). Second, the error contribution from the model vs. observed ΔpN2O comparison is low, with confidence intervals of 0.3 Tg N y −1 for both submodels. Third, ΔpN2O is equally well modelled above the low O2 regions as in the rest of the ocean (Fig. 10, 12), and the contribution of the coastal and deep offshore ocean are nearly proportional to their surface areas (Table 2). These three features are supporting evidence for our results that suggest that the low O2 regions make a small contribution to the global ocean N2O production. They should be balanced against the model bias of the vertical distribution of N2O concentrations, which suggests a larger contribution from the low O2 regions. Freing et al. (2012) also estimated a small fraction of 7% of the global total contributed by denitrification / low O2 N2O production."

We have added a calculation of the contribution of coastal seas, the deep offshore and East equatorial Pacific oceans to N2O flux in Section 3.2 and the Discussion (Table 2 and associated text). And we have expanded on the information about coastal seas and estuaries at the end of the discussion:
"The largest coastal seas are resolved in our model although the processes related to specific coastal environments are not, such as the interactions with sediments and with tides. Our results do not include emissions from estuaries."

- Line 43: The reference to Klawonn et al., 2015 is missing.

The reference was added.

- Line 95, equation 2: More information should be given on this equation, and how it was used in the model/observation comparison. Does using this equation mean that the N2O flux is calculated for a specific period, or that it varies in time? This is unclear.

We have clarified that the model/observations comparison is done at places and months where there are observations. See reply to Gianna Battaglia's comment on Line 199.

Also, there number of significant digits in the various coefficients is way larger than any believable uncertainty associated with the measurements the equations should fit.

We reduced the number of significant digits in Eq. 1. See reply to the same comment on Eq. 1 by reviewer 1.

- Section 2.4, Table 1. Maybe some effort can be done to evaluate the improvements associated withe each model: by adding terms the cost function decreases minimally - is the improvement significant? Does it justify the increase in the model degrees of freedom?

We have used model representations that have relatively few parameters (=degrees of freedom), because the observational data that has been synthesised on a global scale cannot constrain more parameters. Because the prognostic model is an explicit part of the model N-cycle processes, the representations of which are independently constrained by additional observations, it actually has one parameter less (4) than the diagnostic model (5). Akaike's Information Criterion (a criterion that quantifies whether models with more degrees of freedom are "justified" by their increased predictive power $AIC = 1/(n_{observations} - n_{parameters}) * \log(RSS) + 2n_{parameters}$) of the prognostic submodel is 5.9 lower than the diagnostic submodel. This is in the range (2-10) where there is more support for the prognostic model, but there is still some support for the diagnostic model (Burnham,K.P., and D. R. Anderson (1998) Model selection and inference, a practical information-theoretic approach. Springer).

- Line 133-134. The equation could be shown.

We have added: "(Eq. 9 in the supplementary material)" and have also added references to the other relevant equations in the supplementary material in the rest of the Materials and Methods.

- Section 2.6. The slopes (of what, with respect to what?) and relationships used for the model should be clarified with equations, and maybe with corresponding figures (e.g. the observational constrains used). Also, what is the range from which the various slopes were drawn in order to run the different model versions for the optimization?

How were they determined? What values were actually used? Finally, there must be concentration thresholds associated to the transitions between different slopes (e.g. O2). How were these thresholds determined? Were they also optimized for?

The equations, optimised ratios, and range of values tested are given in the supplementary material, we have added references to the relevant equations in Sections 2.4 – 2.6.

- Line 211-212. The reasoning is unclear: an increase in outgassing for a given atmospheric concentration should be driven by a parallel increase in surface concentrations, since the flux is proportional to the concentration (or pN2O) difference. For example, in the limit of removing the saturation N2O concentration, a doubling of the interior production of N2O should double both the outgassing and the surface concentration.

No, a doubling of production leads to a doubling of ΔpN2O, but ΔpN2O/pN2O is small in most of the surface ocean, and the surface concentration increase is proportional to pN2O, not ΔpN2O, so we are correct in stating that a doubling in production leads to only a small increase in surface N2O concentration. We have added this clarification to the manuscript.

- Lines 242-247. This entire paragraph is very unclear, please clarify.

We have clarified this paragraph:

"further observational constraints could not only reduce the error, but also further our understanding of the whole N cycle, including the option of evaluating their model representation against observations, and not just the part that N2O plays in them. Such further constraints are also likely to provide the most productive way to reduce unexplained variability that is found in the observations but not in the present models. E.g., we have shown that both the N2O and NO3 are underestimated at ~300 - 1500 m depth and overestimated below ~2000 m (Fig. 6, 7). Thus, improved representation of mesopelagic remineralisation might lead in improved representation of the N2O depth distribution. However, this falls outside the scope of this study."

- Lines 270-271. Constraining remineralization backwards from N2O production seems a bit far-fetched, given how hard it is to even constrain processes like denitrification alone.

Our point is that the current lack of constraints is not cast in stone. Addressing questions concerning the nitrogen cycle from different angles and integrating the different sources of information in a falsifiable model is more robust than constraining it from the more usual angles of export and nutrient concentrations alone. We added to the end of this paragraph: "Although there are relatively few N2O concentration observations, nitrification and denitrification respond to specific environmental queues (in particular O2 concentration), so that they could contribute a relatively large observational constraint over the full range of environmental conditions."

- Lines 279-281. Please clarify.

See reply to question about Eq. 4 above: Because the paper we discussed only tested sample sizes that were more than 2 orders of magnitude smaller than our database, we decided to delete this paragraph from the discussion.

- Lines 294-297. The issue of biases in model circulation could be assessed by using ventilation tracers, e.g. CFCs. Are they available for this mode?

We are currently including CFCs in our model but this will require time for the development, tuning and validation. The results will not be available for the current study but will inform follow up developments.

- Line 308: do the Authors really think their model can capture costal N2O dynamic, and the massive air-sea fluxes observed there (see Arevalo Martinez et al., 2015),especially in eastern boundary upwellings?

We have added separate analysis of the main N2O hotspots: coastal seas, deep offshore, and East equatorial Pacific oceans. This analysis shows that our two submodels are able to reproduce the observations (see in particular the close correspondence between both submodels and the observations in the high end tail in Fig. 12). Arevalo-Martinez et al. (2015) use the mean N2O flux to represent the whole Peruvian upwelling region. This is similar to linear interpolation with correlation length-scales of the whole region and the whole year. Since their plots suggest that the N2O fluxes are not linearly distributed, this could lead to overestimation of the N2O flux. Therefore we believe our mechanistic model is much more likely to capture realistic N2O dynamics, including in the hotspots, than previously published estimates. For further details see replies to reviewer 2's questions on L32 and L282.

---

## Referee Report (RR1)

I believe that the authors have done a great job revising the manuscript. I have only a couple of technical comments below.

1. P. 3 L. 21: The equation is mathematically incorrect. If an error term is included on the right-hand side, then the appropriate error terms have to be included on the left-hand side too.
2. P. 5 L. 12: in simulate $O_2$ fields → in simulated $O_2$ fields
3. P. 8 L. 32: $_2O$ flux → $O_2$ flux

---

## Author Response (AR2)

Reviewer comments in black, author replies in green.

**Reviewer 1:**

I believe that the authors have done a great job revising the manuscript. I have only a couple of technical comments below.

1. P. 3 L. 21: The equation is mathematically incorrect. If an error term is included on the right-hand side, then the appropriate error terms have to be included on the left-hand side too. We have included the error terms on the left.

2. P. 5 L. 12: in simulate O2 fields  $\rightarrow$  in simulated O2 fields Changed.

3. P. 8 L. 32: 2O flux  $\rightarrow$  O2 flux This was corrected to N2O flux.

**Reviewer 2:**

This paper includes some valuable calculations and the authors have made some good progress toward clarifying their methodology. However, some of the methodological details remain difficult to follow. It also appears that the authors have used only a small subset of the available pN2O data in MEMENTO, for reasons that aren't clear. This subset appears to have a summertime bias in the South Atlantic and a wintertime bias in the North Atlantic. Since pN2O is the one of the main constraints used in the model optimization, these seasonal biases may affect the results.

Please see our point-by-point reply below.

I support publication of this paper in principle, but before it is ready I still think there are a number of details that should be clarified and sections of text that could be written more clearly.

Specific comments

p2L32. The text here mentions 4 methods. Are these 4 enumerated in the lines that follow? For example, is the following method 1,"We extend the global ocean biogeochemistry model PlankTOM10 with additional N cycle processes."? Or are these additional approaches (it seems like there are more than 4 total)? Please start the sentences with transition words like, "First", "Second" instead of just "We" to make this clear. Is the sentence on line35 starting with "Then" approach # 3? If so, please use "Third" This is a key paragraph where the authors lay out what the paper will do, yet I am already lost.

We do not mention 4 methods, but "4 observational databases". These databases are presented in Section 2.2, where they are enumerated ((1), (2), (3), (4)). We have clarified the text from p2L32 onwards, to link the modifications of the model to these four databases. We hope this clarifies our methodology.

P3L30/P4L1. There are a lot more deep and surface N2O data in MEMENTO than the reported n=8047 and n=6136 mentioned here. Were only a subset of the available data selected and why? Also, were the surface data generally in units of ppb and the deep data in nmol/L?

We have used all the data available in the MEMENTO database (on the download dates mentioned) for our analysis. Line 26 states "The number of datapoints reported for each database are after gridding to  $1^{\circ} \times 1^{\circ} \times 12$  months  $\times 33$  depths (World Ocean Atlas 2009)." It is true that the MEMENTO database contains more individual measurements, but these are often taken at high spatial and/or temporal resolution, that do not provide additional constraints on the results. We have added the original number of data points. We have added units to the description of the 4 databases.

P4L4 Are deep data converted to delta\_pN2O or just the surface data? If deep data are converted too, please mention that this necessitates first converting the deep nM data to ppb using a solubility function. This is a large uncertainty. The statement "we have taken the database at face value" is inadequate for conveying the extent of the uncertainties involved in combining nM and ppb data in equation 3.

We did not convert deep data to delta\_pN2O, as the analyses involving measurements in ppb (i.e., air-sea flux estimation in section 3.2) is conducted separately from those involving the deep data in nM (i.e., the analyses constraining N2O production in low O2 regions in Section 3.1). We have clarified this at the end of the introduction and it is repeated in Sections 3.1 and 3.2. See also the next question.

P.4 Section 2.4 (Nitrification) and p5 Section 2.5 (NH4+ uptake) appear to be databases 1 and 2 of the 4 mentioned. I was expecting 2.6 then to be MEMENTO deep N2O and 2.7 to be MEMENTO surface data. Instead, we jump to 2.6 N2O production. This is an example of why it remains challenging to follow the methodology of this paper.

We have clarified which database is used in which section, stating at the end of the introduction that database (1) is used in Section 2.4, database (2) in Section 2.5, database (3) in Section 3.1 and database (4) in Section 3.2. This is because databases (1) and (2) are used for model development while databases (3) and (4) are used for constraining the N2O budget.

P4L27. Does Yool provide an actual data base, or simply assume a constant rate of 0.2 /day everywhere (as written, the sentence implies the latter)?

Both. Yool provided the database of nitrification rate which we use in our analysis. In the same paper, Yool also published model results where they use a constant rate of 0.2/day which we use as the departure point for our analysis. We clarified the text in section 2.4.

Section 2.4. Given that a cost function of 2 means that, on average, the model deviates from the observations by a factor 2 (Section 2.3), does this paragraph suggest that the model deviates from observations by a factor of > 4 on average? In other words, it provides no real constraint. Shouldn't that be stated explicitly somewhere in this section? Is there any meaningful difference between a cost function of 4.22 and 4.16?

Indeed a cost function of >4 is not very satisfactory, though it is not unusual when confronting model results with noisy databases, such as those of ecosystem variables. We modified the text to stress that the differences between 4.22 and 4.16 are minimal. We acknowledge the weak constraints of this database in multiple places in our manuscript. In Section 2.4, by using such phrases "observed nitrification rates are highly variable", "poorly constrains the temperature dependence of AOA", "a slightly improved representation of the observations", "which limits the range of O2 concentrations", "reflecting a lack of data to parameterise an expected decrease", "this estimate is not well constrained". The weak constraints of the data are further stressed in the Discussion: "This lack of data synthesis and of identification of the model to model observed nitrification rates", and "This lack of

knowledge also means that partitioning the global marine N2O production over the nitrification and denitrification pathways is poorly constrained". Finally, we recognised further the weak constraint by omitting our estimate of the low O2 N2O production from the abstract.

P5L34. So, the diagnostic model dN2O/AOU ratios are not optimized against the MEMENTO database using the cost function? The intro and the mention of the 4 datasets had led me to expect they would be.

This is indeed ambiguous. We have added an explanation and clarification to the text: "Previous studies using regional databases have found different oxic ratios (Suntharalingam and Sarmiento 2000 and references therein). Therefore, both the oxic and hypoxic ratios have been reoptimised to the global databases (Sect. 3.1 - 3.2)."

P6L11-12 "We independently optimised the ratios of N2O production and consumption from denitrification" These are minor terms in the budget compared to N2O production from nitrification. Why wasn't that ratio/coefficient optimized?

Both databases were used to optimise separate parameters. To clarify this, P6L14 was rewritten: "The low O2 ratios of both submodels (supplementary material Section 8.7) were optimised using the database of observed N2O concentration (Sect. 3.1) and the oxic ratios of both submodels were optimised using the database of observed DeltapN2O (Sect. 3.2)".

P6L14 "The ratios of both submodels were optimized using the databases of observed N2O concentration and pN2O" Is this referring to the deep N2O or the surface N2O data or both? Both databases were used to optimise separate parameters. To clarify this, it was rewritten: "The low O2 ratios of both submodels (supplementary material Section 8.7) were optimised using the database of observed N2O concentration (Sect. 3.1) and the oxic ratios of both submodels were optimised using the database of observed DeltapN2O (Sect. 3.2)".

Figure 3 caption. Please specify where these values are from (model, MEMENTO, etc). Section 2.1 states: "The globally averaged  $\Delta$ N2O/AOU ratio was calculated from the MEMENTO database (Bange et al., 2009) as  $81.5 \pm 1.4 \mu$ mol/mol (Fig. 3)". This information was moved from the caption to the main body of the text at the previous request of this reviewer (reviewer 2, L72).

Figure 3 annotation shows 0.0815 + 2.7551, but this is reported on p3L15 as  $81.5 \pm 1.4 \mu$ mol/mol. First, the error has completely changed. Second, there is a switch from nmol N2O and umol AOU in the figure to units of umol/mol in the text, with a factor of 1000 thrown in to add to the confusion. Better to be consistent across figure and text. The annotation shows slope and intercept (0.0815x + 2.7551), not the standard deviation of the slope. The units in the text were changed to have mol rather than  $\mu$ mol in the denominator following the previous request of this reviewer (reviewer 2, L262). We note that changing the axes in Fig. 3 to mol O2 and  $\mu$ mol N2O would make the numbers on the axes run from - 0.0003 to 0.0004 and -0.00005 to 0.0002 and therefore difficult to read.

Figure 6/Section 2.5. The model is credited with reproducing "the large scale pattern of surface NH+4 concentration (which) shows an increase with latitude." However, the performance seems pretty poor and the pattern could equally well be described as high in the Southern Ocean (where nutrient utilization is known to be low) and around continental boundaries where there is nutrient input from land. Similar to Section 2.4, the cost function of 3 seems quite large and suggests there's no real constraint here.

The description of the uncertainty was toned down at the previous request of this reviewer (reviewer 2, L144). The text immediately preceding that quoted by the reviewer here still acknowledged the highly variable individual observations and the shortcomings in the model. We have rephrased the latter to clarify it: "the model produces a much smoother distribution of NH4 concentrations than the observations". Our statement on the large-scale patterns was motivated by the comparisons demonstrated in Fig. 6, in particular the zonal average in Fig. 6C, which show an increase with latitude in both hemispheres.

Figure 9 and 11 caption and Equation 5 on p. 6. What does MSE stand for? Please spell it out in all these places. (The captions should be understandable without referring back to the text.) We have added to Section 2.8 after Eq. 5 that: "MSE is mean square error:"

Section 2.8 Should Equation 6 be presented before 5? It seems like 5 builds upon the definition of MSE introduced in Eq 6.

In Eq. 2 and 4 we use the same order of presentation as in Eq. 5, with the equation stating what we want to calculate first, followed by clarifications of the form "in which …".

P7L26. I'm confused by the use of "even though" here. Given that the prognostic model represents N2O consumption in the OMZ, why would that be expected to improve (i.e., increase) the concentration of N2O between 200 and 1500m?

This was clarified to: "even though the prognostic model is more detailed, separately representing the processes of N2O production and consumption at low O2 concentrations.".

P8L5-7. These sentences belong in the methodology. Also, as mentioned above, why isn't the nitrification N2O/NO3 ratio optimized too? That seems like the most important term in the model.

This was moved to the methodology. We clarified p6L14: "the oxic ratios of both submodels were optimised using the database of observed DeltapN2O (Sect. 3.2)". Section 3.2 states: "In the prognostic model, the optimised oxic nitrification ratio was 123  $\mu$ mol N2O (mol NH4+)".

P8L14 I would suggest writing as 0.183 mmol N2O, to avoid switching units, which is confusing for the reader.

We have used the unit of  $\mu$ mol N2O here to enable consistency with our discussion in the following section (3.2).

P8L16-17. "pN2O provided a better constraint than the N2O concentration distribution" Back in Section 2.2, deep N2O is mentioned as dataset number 3 used to optimize the fluxes. The sentence just cited suggests that deep N2O is not actually used. Please clarify. Both databases were used to optimise separate parameters. To clarify this, P6L14 was rewritten: "The low O2 ratios of both submodels (supplementary material Section 8.7) were optimised using the database of observed N2O concentration (Sect. 3.1) and the oxic ratios of both submodels were optimised using the database of observed DeltapN2O (Sect. 3.2)".

Figure 10a) This figure represents only 6136 data points (I think there are a factor of 10 more surface N2O data in MEMENTO than that), yet the figure suggests extensive coverage of the global ocean, and near complete coverage in the Atlantic. Have the data been binned and gridded and if so how? It seems like a single data point has been expanded as a ~5x5 pixel, which implies much better coverage than may really exist. Also, what is the seasonal distribution of the data? I suspect the South Atlantic data are all from austral summer, while

the North Atlantic data are mainly from fall/winter. I don't think the Atlantic is undersaturated to the extent implied by this figure on an annual basis. Can the data be binned by season and plotted in a 4 panel plot?

For gridding see P3L26. We added to the Fig.6 and 10 legend that "observations (symbol size is  $5 \times 5^{\circ}$ )".

The model was subsampled in the same months as the observations, so any seasonal sampling bias in the observations would be reproduced in the model. Because of this, and because the paper already has 12 figures, we have not included the seasonal distribution in the paper, but reproduce it below, showing that (1) no, South Atlantic data is not limited to austral summer, (2) no, North Atlantic data are not limited to fall/winter, but (3) yes, the undersaturation in the North Atlantic is mostly limited to fall (there is no data North of 4°N in the North Atlantic in winter).

---

## Author Response (AR3)

Dear Editor:

Please find our point-by-point responses to the comments by the reviewer below. We feel that we have provided responses already to most of the comments from this reviewer in previous iterations of our paper. We provide below further details and justifications and hope this will satisfy the requests of this journal for publication without further review.

Erik Buitenhuis on behalf of the author team

Reviewer comments in black, author replies in green.

The paper is much improved, although I still find that the description of the optimization of N2O production via nitrification in well-oxygenated waters is somewhat unclear. However, I'm ready to support publication with a few minor revisions.

Specific comments

p2L27-. This paragraph, in which the authors lay out what the paper will do, is greatly improved and now provides a clear blueprint of what to expect.

P3L13 uses => used

changed

P4L2-3 It was very useful to clarify that the 227463 raw data points reduce to only 6136 on a 1x1x12 grid. By my calculations, assuming that 60-70% of grid cells are ice free ocean, this means that only 6136/(360x180x12*~0.65) = ~1% of possible monthly 1x1 grid cells have an N2O measurement. I think this lack of coverage should be mentioned somewhere, perhaps near the presentation of Figure 10b, which conveys the impression of extensive coverage, especially in the Atlantic Ocean. (It is good that the Fig 10 caption mentions the 5x5 pixels, but I think the lack of coverage needs to be acknowledged more explicitly.)

We added a final paragraph to the manuscript to suggest how the N2O flux estimate could be improved. We do not think better coverage of DeltapN2O would significantly help with those:

"To improve the estimate of the ocean N2O flux, first, the uncertainty in the piston velocity would need to be reduced. Once that is achieved, further improvements might be possible by a more accurate model representations of the remineralisation length scale and of the physiology of N2O producing picoheterotrophs (nitrifying and denitrifying Archaea and Bacteria)."

P6L16 indenpendently misspelled

changed

Figure 9. Please explain in the caption what the different symbols and values are. Are these consumption rates? Please give units.

We added "(see legend for a description of the symbols, Tg N y$^{-1}$)"

P8 Section 3. An ongoing point of confusion for me is why the Results only mention the N2O production at low O2. What about the N2O production at higher O2 and total oceanic N2O production? I guess this is explained on p8 29-33, but perhaps make the point more clearly by saying something like, "We used the surface ΔpN2O distribution to constrain N2O production via

nitrification in well-oxygenated waters and thus (by summing with the N2O production at low O2 described in Section 3.1), the total global N2O flux, …"

We added "We ran a range of simulations in which both the (net) low O2 and the oxic N2O production rates were optimised in both submodels (Fig. 10)."

(Because of this addition, what used to be Fig. 11 is now mentioned before Fig. 10, and has been renumbered to become Fig. 10)

P9 paragraph 2. It's not clear what the point of this paragraph is. Is it to argue that coastal areas are not strong sources of N2O? The MEMENTO surface pN2O dataset includes few data in the coastal region, so the calculations in Table 2 may not be well constrained.

The point of the paragraph is to calculate the contribution of potential N2O hotspots. Yes, we do state that there are relatively fewer data in the coastal areas. We also state that the relative constraints are weaker. However, this is relative to an areal flux that is smaller than average, multiplied by a small area, so the absolute contribution to both the globally summed flux and to its uncertainty is small.

P9L27 it's => its

changed

Fig. 10 I still find it very confusing to plot the model results as annual averages in general but as the same month as observations where there are data (what happens if there are obs in 2 or more months in a given grid? Is an average of those months plotted?) For this reason, I recommend including the sets of 4 panel plots provided in the review response in the Supplementary Information. Also, based on those 4 panel plots, there appear to be strong summertime maxima in model dpN2O in both hemispheres. Please comment on whether this is due to enhanced production in summer or simply to thermal effects.

Yes, where there is more than 1 observation, both the observations in panel A and the submodels in panels B and C show the averages of those months. The reviewer does not discount the arguments we already made in our previous reply that: 1) we are analysing a global flux, 2) all data-model differences are included in our analysis and are shown both in Fig. 11 and in a different format in Fig. 12, 3) the piston velocity rather than the model-data mismatch is the main contributor to uncertainty. Therefore we still maintain these figures would not add materially to the paper.

P9 last paragraph. Please provide more information about Cohen and Gordon's calculation. How was it based on N assimilation? What was their total estimate?

We added their total estimate "as 4 – 10 Tg N y$^{-1}$". We follow completely standard procedure in providing a reference rather than repeating all the details of their calculations.

P10L4 "nitrification, which uses O2 as the electron acceptor."

added

P10L5 "needed to nitrify NH4+ to NO3-, the electron acceptor"

added

P10L21 Is N really in the 0 and +2 states in N2O? I thought both Ns were in the +1 state.

N2O is an asymmetrical molecule, the middle N that is bonded to the O is different from the N that is only bonded to the middle N. The answer appears to be ambiguous, though. One source states the 0/+2 configuration based on a triple NN bond and a single NO bond. Based on double bonds between both NN and NO one would get -1/+3. Yet another source states that based on synthesis of N2O by double dehydration of NH4 and NO3 it is -3/+5. Since the indicated states are not definitely wrong, we've left them.

P10L35/P11L1. This claim is not obviously supported by Figures 10-12. The fact that the depth profiles are significantly off seems like a red flag that low O2 production is underestimated, given how sparse the surface dpN2O data are and their sensitivity to air-sea transfer assumptions.

Fig. 12 quite clearly shows that the bulk of the model-data mismatches occurs at low ΔpN2O, so if anything our statement that "ΔpN2O is equally well modelled above low O2 region" (i.e., where ΔpN2O is high) is an understatement. And no, we also show in Table 2 that ΔpN2O are not sparse in the N2O hotspots. We added a final paragraph to the manuscript to reiterate the sensitivity to the air-sea transfer function (=the piston velocity): "To improve the estimate of the ocean N2O flux, first, the uncertainty in the piston velocity would need to be reduced. …"

P12 Paragraph 2. This is mainly a repeat of the previous paragraph

Indeed. The second version was deleted.

The reviewers gave the same response to 2 of my previous comments.

My review #2 comment was: P6L11-12 "We indenpendently optimised the ratios of N2O production and consumption from denitrification" These are minor terms in the budget compared to N2O production from nitrification. Why wasn't that ratio/coefficient optimized?

The authors responded: Both databases were used to optimise separate parameters. To clarify this, P6L14 was rewritten: "The low O2 ratios of both submodels (supplementary material Section 8.7) were optimised using the database of observed N2O concentration (Sect. 3.1) and the oxic ratios of both submodels were optimised using the database of observed DeltapN2O (Sect. 3.2)".

However, this does not really address my comment. Furthermore, they gave a nearly identical response to my next comment: P6L14 "The ratios of both submodels were optimized using the databases of observed N2O concentration and pN2O" Is this referring to the deep N2O or the surface N2O data or both?

The authors responded: Both databases were used to optimise separate parameters. To clarify this, P6L14 was rewritten: "The low O2 ratios of both submodels (supplementary material Section 8.7) were optimised using the database of observed N2O concentration (Sect. 3.1) and the oxic ratios of both submodels were optimised using the database of observed DeltapN2O (Sect. 3.2)".

I think there was a typo in which the authors pasted the same response to 2 of my comments, without actually addressing the first comment

We did address the first comment. The first comment asked why the nitrification ratio wasn't optimised. Our response explicitly added to the manuscript that the oxic ratios were optimised. We state oxic ratio rather than nitrification ratio because the former applies to both submodels and the latter only to the prognostic model. The structure of the prognostic model is explained in the second paragraph of Section 2.6, where it is stated on page 6, line 13 that nitrification is oxic. (The remainder of the paragraph makes it clear it's the only oxic pathway.)

[revised manuscript text omitted]

**Figure 5.** N$_2$O yield of nitrification (N atom:atom) as a function of O$_2$ concentration, filled triangles: AOA (**?**), open circles: AOB at low to medium cell numbers (**??**), crosses: marine AOB at high cell numbers (**??**), plusses: soil AOB at high cell numbers (**?**). Black line: logarithmic fit to AOA and low to medium cell number AOB (yield = 0.791-0.126·ln(O$_2$) mmol N in N$_2$O (mol NH$_4^+$)$^{-1}$).

[Figure]

**Figure 6.** Surface NH$_4^+$ concentration ($\mu$mol L$^{-1}$). A) observations (symbol size is 5 × 5°). B) model results are for the same months where there are observations, and annual averages everywhere else. C) zonal average, black) observations, red) model results. Model results are for the same months and longitudes as the observations. Latitude y-axis to the left of panel A.

[Figure]

**Figure 7.** Depth profiles of $N_2O$ concentration (nmol $L^{-1}$) for different basins. Black lines: observations, Green lines: optimised diagnostic model, Red lines: optimised prognostic model.

[Figure]

**Figure 8.** Depth (m.) profile of average $NO_3^-$ concentration ($\mu$mol $L^{-1}$). Black line) WOA2009 synthesis of observations, not interpolated. Red line) Model results sampled at the places where there are observations.

[Figure]

**Figure 9.** $MSE^{0.5}$ for the two $N_2O$ submodels compared to the $N_2O$ concentration database as a function $N_2O$ production in the low $O_2$ regions. $MSE_{min}$ was obtained as the minimum of a second order polynomial fit (black lines). The $1\sigma$ confidence interval, where MSE equals the value calculated from Eq. 5, is indicated by the horizontal lines. A) diagnostic submodel, each point represents a simulation with a different low $O_2$ ratio, B) prognostic model, "no c" is with no $N_2O$ consumption i.e. net production = gross production. All other lines have a constant gross production (see legend for a description of the symbols, Tg N $y^{-1}$), and net production varies with different $N_2O$ consumption rates. Range of parameter values is given in the supplementary material Section 8.7.

[Figure]

**Figure 10.** $MSE^{0.5}$ for the two $N_2O$ submodels compared to the $\Delta pN_2O$ database as a function of global $N_2O$ flux at different (net) $N_2O$ production rates in the low $O_2$ regions. $MSE_{min}$ and confidence intervals as in Fig. 8. A) diagnostic submodel, the four lines represent the four best low $O_2$ production rates from Fig. 9A, each point represents a simulation, different symbols indicate different low $O_2$ ratios, points with the same symbols have different oxic $N_2O$ production ratios. B) prognostic submodel, the four lines represent the optimised net production rates at the four best gross production rates from Fig 9B, points with the same symbols have different $N_2O$ ratios for nitrification.

[Figure]

**Figure 11.** Surface $\Delta pN_2O$ (ppb). A) observations (symbol size is $5 \times 5°$), B) optimised diagnostic model, C) optimised prognostic model. Model results are for the same months where there are observations, and annual averages everywhere else. D) zonal average, Black line: observations, Green dashed: diagnostic model, Red dotted: prognostic model. Model results are for the same months and longitudes as the observations. Latitude y-axis to the left of panel A.

[Figure]

**Figure 12.** Frequency distribution of $\Delta pN_2O$ in the observations (solid black), and the optimised simulations of the diagnostic submodel (green squares) and the prognostic submodel (red lines).